# BTBS-LNS: A Binarized-Tightening, Branch and Search Approach of Learning Large Neighborhood Search Policies for MIP

## ABSTRACT

Learning to solve the large-scale Mixed Integer Program (MIP) problems is an emerging research topic, and policy learning-based Large Neighborhood Search (LNS) has recently shown its effectiveness. However, prevailing approaches predominantly concentrated on binary variables and often susceptible to becoming ensnared in local optima derived from the learning complexity. In response to these challenges, we introduce a novel technique, termed **B**inarized-**T**ightening **B**ranch-and-**S**earch **LNS** (**BTBS-LNS**). Specifically, we propose the "Binarized Tightening" technique for integer variables to deal with their wide range by encoding and bound tightening, and design an attention-based tripartite graph to capture global correlations within MIP instances. Furthermore, we devised an extra branching network at each step, to identify and optimize some wrongly-fixed backdoor variables[1] by the learned LNS policy. We empirically show that our approach can effectively escape local optimum in some cases. Extensive experiments on different problems, including instances from Mixed Integer Programming Library (MIPLIB), show that it significantly outperforms the open-source solver SCIP and LNS baselines. It performs competitively with, and sometimes even better than the commercial solver Gurobi (v9.5.0), especially at an early stage. Source code will be made publicly available.

## 1 INTRODUCTION AND RELATED WORK

Mixed-integer programming (MIP) is a well-established and general optimization problem and has been widely studied across applications. In many cases, feasible or even optimal solutions are required under strong time limits, and thus efficiently finding high-quality solutions is of great importance in real-world scenarios. Recently, machine learning for combinatorial optimization has been an emerging topic (Bengio et al., 2021) with prominent success in different tasks, e.g. graph matching (Yan et al., 2020), and ML4MIP is also an emerging field (Zhang et al., 2023).

A variety of deep learning based solving methods were proposed to deal with specific MIP problems, including construction methods (Ma et al., 2019; Xing & Tu, 2020; Fu et al., 2021; Zhang et al., 2020; Khalil et al., 2017; Xin et al., 2021) and iterative based refinements (Wu et al., 2021b; Chen & Tian, 2019; Lu et al., 2019b; Li et al., 2020). While they cannot be directly applied to a wider scope of MIP problems, and thus learning the solving policies for general MIP problems has been also intensively studied, in which the primal heuristics catch more attention, including Large Neighborhood Search (LNS) (Wu et al., 2021a; Song et al., 2020; Nair et al., 2020a) and Local Branching (LB) (Liu et al., 2022). In this paper, we focus on LNS for solving general MIP problems – the most powerful yet also the most expensive iteration-based heuristics (Hendel, 2022).

Traditional LNS methods usually explore a complex neighborhood by predefined heuristics (Gendreau et al., 2010; Altner et al., 2000; Godard et al., 2005; Yagiura et al., 2006; Lee, 2009), in which the heuristic selection is a long-standing challenging task, especially for general MIP problems, which may require heavy efforts to design valid heuristics. Learning-based methods provide a possible direction. For example, both Imitation Learning (IL) (Song et al., 2020) and Reinforcement

---

[1]In this paper, we extend the meaning of backdoor variables Williams et al. (2003) to those with different solutions compared with global optimum.

Learning (RL) (Wu et al., 2021a; Nair et al., 2020a) showed effectiveness in learning decomposition-based LNS policies. While there still exist some challenges. The performance of the learned policies may significantly degrade when applied to general integers due to the vast scale of candidate values (compared to binary variables), leading to a large complexity in optimization. Moreover, the learned policies may be trapped in local optimum when dealing with some complicated cases.

In this paper, we propose a Binarized-Tightening, Branch and Search based LNS approach (**BTBS-LNS**) for general MIP problems. Specifically, we design the "Binarized Tightening" algorithm to deal with the optimization for general integer variables. In particular, we first binarize the general integer variables and express them with the resulting bit sequence, and then tighten the bound of original variables w.r.t. the LNS decision along with the current solution. In this way, the variable bounds can be tightened and effectively explored at a controlled complexity. Based on our binarization formulation, we further develop an attention-based tripartite graph (Ding et al., 2020) to encode the MIP instances with three types of nodes, including objectives, variables, and constraints, which delivers better expression. Meanwhile, to enhance exploration and optimize some wrongly-fixed backdoor variables (Khalil et al., 2022) by the learned LNS policy, we leverage an extra branching graph network at each step, providing branching decisions at global (or local) view to help escape local optimum. In a nutshell, this paper can be characterized by the following bullets:

**1) Bound Tightening for MIP.** We propose the "Binarized Tightening" scheme for general MIP problems with an efficient embodiment of variable encoding and bound tightening techniques.

**2) Problem encoding with attentional tripartite graph.** We develop an attention-based tripartite graph to encode MIP problems, showing the potential to learn valid representations for general MIP.

**3) Combining LNS with branching.** We devise a variable branching mechanism to select and optimize the wrongly-fixed backdoor variables by the learned LNS policy at each step. The hybrid branch and search policy greatly enhances exploration and shows efficiency.

**4) Strong empirical results.** Experiments on seven MIP problems show that our method consistently outperforms the LNS baselines and open-source SCIP (Gamrath et al., 2020). In some cases, it even achieves superior performance over Gurobi, purely taking SCIP as the baseline solver. It can further boost Gurobi when taking Gurobi as the baseline solver. The source code will be released.

We elaborate on the detailed comparison with existing works in Appendix A.1.

## 2 PRELIMINARIES

We introduce MIP and its mainstream solving heuristic: Large Neighborhood Search (LNS).

**Mixed Integer Program (MIP)** is in general defined as:
$$\begin{aligned} \min \quad & \mathbf{c}^\top \mathbf{x} \\ s.t. \quad & \mathbf{A}\mathbf{x} \leq \mathbf{b} \\ & x_i \in \{0,1\}, \forall i \in \mathcal{B}; x_j \in Z^+, \forall j \in \mathcal{G}; x_k \geq 0, \forall k \in \mathcal{C} \end{aligned} \tag{1}$$
where $\mathbf{x} \in \mathbb{R}^n$ is a vector of $n$ decision variables; $\mathbf{c} \in \mathbb{R}^n$ denotes the vector of objective coefficients. $\mathbf{A}\mathbf{x} \leq \mathbf{b}$ denotes the overall $m$ linear constraints, where $\mathbf{A} \in \mathbb{R}^{m \times n}$ represents the incidence matrix, with $\mathbf{b} \in \mathbb{R}^m$. For general MIP instances, the index set of $n$ variables $\mathcal{N} := \{1, ..., n\}$ can be partitioned into three sets, binary variable set $\mathcal{B}$, general integer variable set $\mathcal{G}$ and continuous variable set $\mathcal{C}$. MIP is more difficult to deal with compared with integer programming Wu et al. (2021a) as the continuous variables may require distinct optimization policies with integer variables.

**Large Neighborhood Search (LNS)** is a powerful yet expensive heuristic for MIP (Gendreau et al., 2010). It takes so-far the best feasible solution $\mathbf{x}^*$ as input and searches for the local optimum in its neighborhood:
$$\mathbf{x}' = \arg \min_{\mathbf{x} \in N(\mathbf{x}^*)} \{\mathbf{c}^\top \mathbf{x}\} \tag{2}$$
where $N(\cdot)$ is a predefined neighborhood, denoting the search scope at each step, and $\mathbf{x}'$ denotes the optimized solution within $N(\mathbf{x}^*)$, obtained by destroying and re-optimization from current solution.

Compared to local search heuristics, LNS can be more effective by using a larger neighborhood. However, the selection of neighborhood function $N(\cdot)$ is nontrivial. Heuristic methods mainly rely on problem-specific operators, e.g., 2-opt (Flood, 1956) in TSP, which call for considerable trial-and-error and domain knowledge (Papadimitriou & Steiglitz, 1998). The popular learning-based

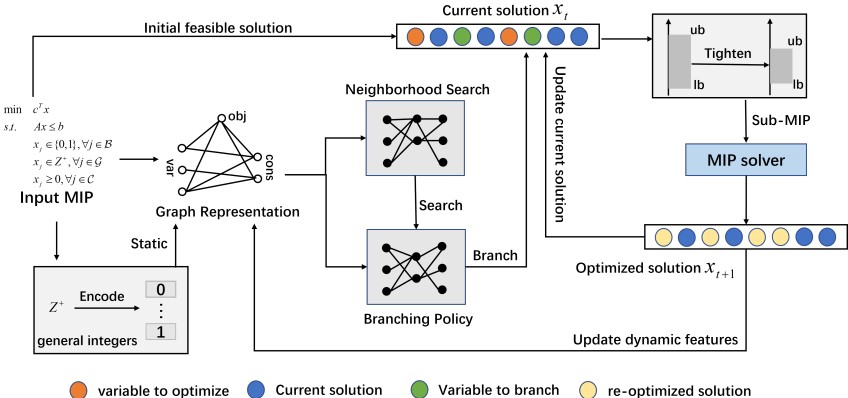

Figure 1: Overview of **BTBS-LNS**. First, we propose "Binarize Tightening" to handle general integer variables. The *Binarize* mechanism can binary-encode the variables and split them into sub-optimization bits. With the bit-wise decision by LNS, the variable upper/lower bounds can be refined by bound tightening. Second, we devise a branching network on top of pure LNS policy to select wrongly-fixed backdoor variables by pure LNS policy, thus making up for its local search limit.

policies mainly focus on binary variables and may be trapped in local optimum in some complicated cases. To obtain a general neighborhood function, we propose a binarized-tightening branch-and-search LNS approach. It destroys, branches, and re-optimizes the initial solution.

## 3 METHODOLOGY

### 3.1 OVERVIEW

Fig. 1 presents the overview of our approach. The input is an MIP instance, with its initial feasible solution $\mathbf{x}_0$ generated by a baseline solver. The general integer variables are firstly encoded to binary substitute variables, and the instance is then represented as a tripartite graph Ding et al. (2020) and fed into the large neighborhood search network, selecting the variable subsets that may need to optimize at each step, with the remaining variables fixed or bound tightening (see Sec. 3.2 and 3.3). Subsequently, we devise an extra branching network to select some wrongly-fixed backdoor variables by the learned LNS policy, to help escape local optimum. With the sequential decisions of the branch and search policy and the resulting tightened variable bounds, an off-the-shelf solver, e.g. SCIP, is applied to obtain the optimized feasible solution $\mathbf{x}_{t+1}$. Iterations continue until the time limit is reached, and the optimized solutions can be finally obtained.

---

**Algorithm 1** Bound tightening for integer variable $x_i$

**Input**: Initial lower and upper bound of $x_i$: $lb$, $ub$;
Current solution value: $x_i = p$;
Binary LNS decision for $x_i$: $a_i^t$ for unbounded variables, and $\{a_{i,j}^t | j = 1, 2, ..., d\}$ for others.
**Output**: Tightened $lb$, $ub$

1: **if** $x_i$ unbounded **then**
2:     **if** $lb$ existed **and** $a_i^t = 0$ **then**
3:         $ub = 2p - lb$
4:     **else if** $ub$ existed **and** $a_i^t = 0$ **then**
5:         $lb = 2p - ub$
6:     **end if**
7: **else**
8:     $d = \lceil \log_2 (ub - lb) \rceil$
9:     **for** $j = 0 : d$ **do**
10:         **if** $a_{i,j}^t = 0$ **then**
11:             $lb = \max(lb, p - 1/2(ub - lb))$;
12:             $ub = \min(ub, p + 1/2(ub - lb))$;
13:         **else**
14:             break;
15:         **end if**
16:     **end for**
17: **end if**

---

In general, the neighborhood search policy and branching policy are trained sequentially, where the training details can refer to Sec. 3.3 and Sec. 3.4, respectively. They optimize the current solution at different view and may remedy the local search drawbacks of the learned LNS policy in some cases.

### 3.2 THE BINARIZED TIGHTENING SCHEME

Variables in general MIP instances can be divided into three categories: binary, general integer (with arbitrary large value), and continuous variables. Previous studies mainly focused on the binary variables $(0/1)$. Limited values greatly simplify the optimization, making it easier to deal with compared to the general integer variables, and some learning frameworks have proved their effectiveness (Wu

et al., 2021a; Song et al., 2020). In this paper, we concentrated on more general MIP problems, especially for general integer variables.

An intuitive method is to directly migrate some efficient binary LNS approaches, e.g., Wu et al. (2021a), to general integers. In this way, different types of variables are equally treated, and at each step, we fix some of the variables (no matter what type the variable belongs to), and solve the sub-MIP with a baseline solver e.g. SCIP (Gamrath et al., 2020) or Gurobi (G., 2020). However, empirical results revealed that the simplified generalized LNS approach is much slower and significantly underperforms the MIP solvers, e.g., Gurobi.

To illustrate the underlying reason, we take a simple MIP instance as example:

$$\begin{aligned} \min \quad & x \\ s.t. \quad & x + y = z \quad x, y \in \{0, 1\}, z \in Z^+ \end{aligned} \tag{3}$$

Assume that the initial feasible solution is $(x, y, z) = (1, 1, 2)$. At a certain iteration when the general integer variable $z$ is fixed, the remaining sub-MIP problem cannot be optimized as $z$ has a strong correlation with all the other variables, making it difficult to deal with by simply fixing.

We propose the so-called "Binarized Tightening" scheme for MIP. The idea is that we tend to confine the variables within a narrow range around the current solution rather than directly fixing them, to balance exploration and exploitation. It shares similar insights with local search, which relies on the current best solution to guide the search, thus avoiding blind search throughout the entire solution space. Specifically, we represent each general integer variable with $d = \lceil \log_2 (ub - lb) \rceil$ binary variables at decreasing significance, where $ub$ and $lb$ are original variable upper and lower bounds, respectively. The subsequent optimization is applied to the substitute binary variables, indicating current solution reliable or not at corresponding significance. In this way, we transform the LNS for the original variable to multiple decisions on substitution variables. Note that the unbounded variables where $ub$ or $lb$ does not exist, will not be encoded and will remain a single variable.

The action for each substitute variable can be obtained from the LNS policy (see Sec. 3.3), where 0 means the variable indicates reliable at current significance, and 1 means it still needs exploration. We design a bound-tightening scheme to fully use the bit-wise action in Alg. 1. Specifically, let $a_{i,j}^t$ represent the action for the $j^{th}$ substitute variable of variable $i$ at step $t$. Actions $a_{i,j}^t$ for all $j$ are checked, and the upper and lower bounds will be updated and tightened around the current solution every time when $a_{i,j}^t = 0$, as in Line 11-12. Therefore, more fixed substitute variables can contribute to tighter bounds. In our embodiment, variables that sit far from both bounds can have a significantly wider exploration scope than close-to-bound variables, as they showed no explicit "preference" on either bound direction, which is significantly different from Nair et al. (2020b) (see Appendix A.1 for detailed discussion). Tightening on either bound when the current solution sits precisely at the midpoint of variable bounds, may contribute to performance degradation derived from reduced exploration, which conceptually drives us to design the bound tightening scheme, tightening the bounds on the far side iteratively..

In addition, as for unbounded variables, our meticulous analysis on MIPLIB benchmark set (Gleixner et al., 2021) revealed that all unbounded variables within the instances are characterized by unbounded in only one direction, which means that either $lb$ or $ub$ will exist for all general integer variables. In this respect, we define a virtual upper (lower) bound when $a_i^t = 0$ as in Line 3 and 5, which share similar insights with regular variables to put the current solution at precisely the midpoint of the updated bounds.

**Connection with bound tightening techniques in constraint programming.** Bound tightening techniques have been commonly applied in some constraint integer programming problems, including Constraint Propagation (Achterberg, 2007; Savelsbergh, 1994; Moskewicz et al., 2001), Optimization-Based Bound Tightening (OBBT) (Gleixner et al., 2017), Feasibility-Based Bound Tightening (FBBT) (Belotti et al., 2012), and so on. They share similar insights with our approach in terms of reducing the solving complexity of the re-defined problem. These techniques aim to maintain optimality, making them sometimes computationally expensive. However, our iterative refinement procedure for bound tightening differs from them. The iterative optimization scheme focuses on searching for better feasible solutions within the neighborhood of the current solution, guided by the learned policy. Consequently, our approach allows for a significant reduction of the complexity of the re-defined problem, leading to improved solutions efficiently.

### 3.3 Graph-based LNS policy parameterization

Bipartite graph is recently popular utilized in Gasse et al. (2019), Nair et al. (2020b), and Wu et al. (2021a) to represent the MIP instance states. However, the objective is not explicitly considered, which may contribute to performance degradation in some cases, e.g., when all discrete variables do not exist in the objectives (Yoon, 2022). To capture the correlations between objectives with variables and constraints reasonably, we propose to describe the input instance as a tripartite graph $\mathcal{G} = (\mathcal{V}, \mathcal{C}, \mathcal{O}, \mathcal{E})$, where $\mathcal{V}$, $\mathcal{C}$, and $\mathcal{O}$ denote the variable, constraint, and objective nodes, and $\mathcal{E}$ denotes the edges. The features of nodes and edges can refer to Appendix A.3, where the new objective node representations are defined as the average states of corresponding variables.

We parameterize the policy $\pi_\theta(a_t|s_t)$ by an attention-based Graph Convolution Network (GCN). Slightly different from graph attention networks (Veličković et al., 2018), we remove the *softmax* normalization to fully reserve the absolute importance between neighborhood nodes and edges, which may help capture the contributions for each node to the final objectives (see Appendix A.7 for detailed comparison). The message passing $\mathcal{C} \to \mathcal{V}$ is as follows (likewise for others):

$$\mathbf{h}_i^{t+1} = f_{\mathcal{CV}} \left( \text{CONCAT} \left( \mathbf{h}_i^t, \frac{\sum\limits_{j \in \mathcal{C} \cap N_i} w_{ij}^t (\mathbf{h}_j^t + \mathbf{h}_{e_{ij}}^t)}{|\mathcal{C} \cap N_i|} \right) \right) \tag{4}$$

where $\mathbf{h}_i^t$ and $\mathbf{h}_{e_{ij}}^t$ denote the features of node $i$ and edge $(i, j)$ at step $t$; $f_{\mathcal{CV}}$ is a 2-layer perceptron with relu activation that maps the current states to the next iteration $\mathbf{h}_i^{t+1}$; $N_i$ denotes the neighborhood nodes of $i$ and $|\mathcal{C} \cap N_i|$ denotes the counts of neighborhood constraint nodes for node $i$, utilized to normalize the weighted sum neighboring features; $w_{ij}^t$ denotes the weighted coefficient between node $i$ and node $j$ at step $t$, measuring their correlations as follows, where $\mathbf{W}_{\mathcal{CV}}$ denotes the weight matrix between constraint and variable nodes.

$$w_{ij}^t = \sigma_s(\mathbf{W}_{\mathcal{CV}} \cdot \text{CONCAT}(\mathbf{h}_i^t, \mathbf{h}_{e_{ij}}^t, \mathbf{h}_j^t)) \tag{5}$$

At each graph attention layer, the message passing between different types of nodes are: $\mathcal{V} \to \mathcal{O}$, $\mathcal{O} \to \mathcal{C}$, $\mathcal{V} \to \mathcal{C}$, $\mathcal{C} \to \mathcal{O}$, $\mathcal{O} \to \mathcal{V}$, $\mathcal{C} \to \mathcal{V}$, which are calculated as Eq. 4 sequentially. In this way, after $K$ iterations (throughout this paper $K = 2$), the features for both the nodes and edges are updated. We finally process the variable nodes by a multi-layer perceptron and the output value can be regarded as the *destroy* probability for each variable at this step, serving as the neighborhood search policy in Fig. 1. The neighborhood search policy was trained with Q-actor-critic algorithm by RL, following the same protocol with Wu et al. (2021a), while with the following major differences:

**States**: We adopt an attentional tripartite graph to capture correlations among variables, constraints, and objectives. Details of the features are gathered in Table 7.

**Actions**: For the general variable $x_i$ represented with $d$ substitutes, the LNS decision at step $t$ will contain $d$ binary actions $a_{i,j}^t$, indicating the current solution reliable or not at current significance (see Alg. 1).

**Transition and rewards**: We follow the same protocol as in Wu et al. (2021a), where the next state $s_{t+1}$ is obtained by the baseline solver, and the reward is defined as objective value improvements.

---

**Algorithm 2** Branch and search at the $t^{th}$ step

**Input**: Number of variables $n$;
LNS decisions $N^t = \{n_i^t | i = 1, 2, ..., n\}$;
branching decisions $B^t = \{b_i^t | i = 1, 2, ..., n\}$;
variable set $\mathbf{x} = \{x_i | i = 1, 2, ..., n\}$;
best solution at the $t^{th}$ step $\mathbf{x}^t = \{x_i^t | i = 1, 2, ..., n\}$;
The ratio for branching variables $r$;
**Output**: $\mathbf{x}^{t+1}$;

1: Let $D = \emptyset$;
2: **while** $i \leq n$ **do**
3:     **if** $x_i$ is general integer variable **then**
4:         Tighten the bound as in Alg. 1 using $n_{i,j}^t$ (with $d$ separate decisions);
5:     **else**
6:         **if** $n_i^t = 0$ **then**
7:             Fix the value $x_i^{t+1} = x_i^t$;
8:         **else**
9:             **if** $b_i^t = 1$ and $x_i$ is binary variable **then**
10:                $D = D \cup \{i\}$;
11:             **end if**
12:         **end if**
13:     **end if**
14: **end while**
15: add constraint $\sum\limits_{i \in D} |x_i^{t+1} - x_i^t| \leq rn$ to sub-MIP;
16: Optimize $\mathbf{x}^{t+1}$ with the solver;

---

### 3.4 BRANCHING POLICY

As discussed above, previous single-policy approaches were easy to be trapped in local optimum at an early stage in some complicated tasks. To remedy this issue, an intuition is to select and optimize those wrongly-fixed backdoor variables by LNS policy at each step. With this insight, we proposed to learn an extra branching network with imitation learning on top of LNS to filter out those variables at each step. Note that it was only applied to binary variables which are more likely to be backdoors that fixed earlier leading to local optima.

The most critical issue for the branching policy learning is the collection of branching variable labels. In other words, we need to figure out how to identify the potentially wrongly-fixed variables at each step. We proposed two different variants, which deals with the issue in global and local view as in Fig. 2:

**Global branching (BTBS-LNS-G):** It gathers labels from the fixed variables by LNS at each step and contrast them with the global optimal solution. Variables that exhibit differing values between these solutions are indicative of potentially misclassified variables within the current LNS decisions from a global perspective. Since the global optimal solution may be too difficult to acquire in a reasonable time for hard instances, it was replaced by the best-known solution in our embodiment.

**Local branching (BTBS-LNS-L):** Different from the global view contrast, it gathers labels by incorporating the following local branching constraints (Liu et al., 2022) at each step:

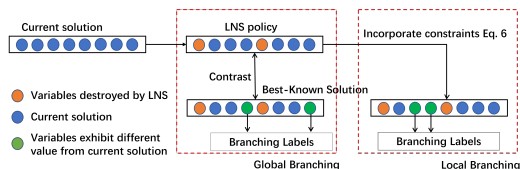

$$\sum_{i \in \mathcal{B} \cap \mathcal{F}} |x_i^{t+1} - x_i^t| \le k \qquad (6)$$

Figure 2: Global branching **vs** Local branching.

where $\mathcal{F}$ denotes the currently fixed variables set by LNS. With this extra constraint, the re-defined sub-MIP problem can be solved by the baseline solver, and up to $k$ changed fixed variables will be selected at a local view as the branching variable labels at current step. The selected variables can be regarded as locally wrongly-fixed variables by LNS.

With the collected labels, the branching network can be offline trained. The inputs are tripartite graph-based features (see Table.= 7 in Appendix A.3 for detail), where we additionally append the LNS decisions made by the learned LNS policy as variable features, as we only focused on the fixed variables for extra branching. Note that the input states are collected by resolving the training instances, along with the learned LNS policy. And the labels are also gathered within the resolving at each step. Then the graph-based features are fed into a similar graph attention network as described in Sec. 3.3 to update the node/edge representations. We finally process the variable nodes by a multi-layer perceptron (MLP) and the output value can be regarded as the branching probability for each variable at this step. Cross-entropy loss was utilized to train the branching network to bring the outputs closer to the collected labels, with the pipeline as Alg. 3 in Appendix A.2.

Note that except for the different label collection scheme, **BTBS-LNS-L** and **BTBS-LNS-G** remain all the same. In general, the learned branching policy takes effect on top of LNS, enhancing exploration and optimizing its wrongly-fixed backdoor variables at each step. The pipeline for the hybrid framework is given in Alg. 2, where we select $r = 10\%$ of the variables with maximum branching probability to branch on at each step in the inference phase. In general, the hybrid branch and search policy work together to formulate the sub-MIP at each step (see Line 15). As can be seen from the experimental results in Table 1 to Table 3, hybrid branch and search clearly outperforms pure LNS policy, even better than the commercial solver Gurobi (G., 2020) in many cases.

## 4 EXPERIMENTS

### 4.1 SETTINGS AND PROTOCOLS

**Peer methods.** We compare with the following baselines, and more details are illustrated in Appendix A.3. All methods are solved in 200s time limit by default.

**1) SCIP (v7.0.3), Gurobi (v9.5.0)**: state-of-the-art open source and commercial solvers, and were fine-tuned with the aggressive mode to focus on improving the objective value.

**2) U-LNS, R-LNS, DINS, GINS, RINS and RENS**: heuristic LNS methods (Achterberg, 2007).

Table 1: Comparison with baselines for binary Integer Programming (IP) with four hard problems: SC, MIS, CA, MC. We also let SCIP run for a longer time (500s with SCIP (500s) and 1000s with SCIP (1000s), respectively). So for Gurobi and our **BTBS-LNS** in other tables.

| Methods | Set Coverin (SC) | | | Maximal Independent Set (MIS) | | | Combinatorial Auction (CA) | | | Maximum Cut (MC) | | |
|---|---|---|---|---|---|---|---|---|---|---|---|---|
| | Obj | Gap% | PI | Obj | Gap% | PI | Obj | Gap% | PI ($\times 10^3$) | Obj | Gap% | PI |
| SCIP | 563.92 | 3.23 | 20225 | -684.52 | 0.25 | 312.25 | -109960 | 4.71 | 3312.4 | -852.64 | 8.01 | 15193 |
| SCIP (500s) | 553.11 | 1.40 | / | -684.98 | 0.18 | / | -111511 | 3.36 | / | -861.55 | 7.11 | / |
| SCIP (1000s) | 551.33 | 1.06 | / | -685.66 | 0.09 | / | -112627 | 2.40 | / | -863.99 | 6.87 | / |
| U-LNS | 567.70 | 3.84 | 22459 | -680.44 | 1.50 | 1145.4 | -104526 | 9.42 | 4003.0 | -865.32 | 6.72 | 11565 |
| R-LNS | 569.40 | 4.17 | 23015 | -682.54 | 1.29 | 693.45 | -107407 | 6.92 | 3631.2 | -868.95 | 6.33 | 10923 |
| FT-LNS | 565.28 | 3.48 | 20988 | -680.84 | 1.42 | 1103.7 | -104048 | 9.83 | 4123.6 | -869.29 | 6.30 | 10554 |
| DINS | 567.88 | 3.97 | 22735 | -682.71 | 1.24 | 657.5 | -108948 | 4.48 | 3337.4 | -872.33 | 5.75 | 10006 |
| GINS | 567.28 | 3.81 | 22197 | -683.24 | 0.75 | 683.6 | -107548 | 6.90 | 3599.8 | -874.62 | 5.41 | 9765.0 |
| RINS | 566.52 | 3.63 | 21835 | -681.75 | 1.32 | 816.5 | -106548 | 7.33 | 3843.4 | -870.17 | 6.04 | 10277 |
| RENS | 561.48 | 2.35 | 19112 | -683.12 | 0.79 | 792.36 | -109025 | 4.40 | 3125.2 | -875.44 | 5.29 | 9116 |
| RL-LNS | 552.38 | 1.29 | 17623 | -685.74 | 0.07 | 182.63 | -112666 | 2.36 | 2271.6 | -888.25 | 4.25 | 6538 |
| Branching | 557.41 | 1.72 | 18007 | -685.70 | 0.07 | 183.44 | -111835 | 3.09 | 2492.7 | -891.58 | 3.99 | 6104 |
| LNS-TG | 548.65 | 0.66 | 16828 | -685.69 | 0.08 | 182.24 | -112711 | 2.32 | 2247.8 | -898.28 | 3.05 | 4782.6 |
| LNS-Branch | 551.55 | 1.11 | 17234 | -685.65 | 0.09 | 182.19 | -112666 | 2.36 | 2275.3 | -891.59 | 3.73 | 5840.0 |
| LNS-ATT | 548.45 | 0.65 | 16714 | -685.75 | 0.07 | 182.10 | -112820 | 2.23 | 2231.5 | -902.11 | 2.99 | 3975.1 |
| **BTBS-LNS-L** | 547.88 | 0.47 | 16234 | -685.86 | 0.05 | 181.47 | -112864 | 2.18 | 2196.8 | -909.17 | 1.99 | 2518 |
| **BTBS-LNS-G** | **547.48** | **0.35** | **16205** | -685.92 | 0.05 | 178.35 | **-113742** | **1.43** | **1998.9** | **-922.18** | **0.59** | **785** |
| Gurobi | 549.44 | 0.75 | 16796 | **-686.24** | **0** | 173.15 | -113731 | 1.44 | 2075.4 | -921.90 | 0.62 | 842 |

Table 2: Generalization to large-scale binary integer programming (IP) instances using the trained polcies from small problems in Sec. 4.2.

| Methods | SC2 | | | MIS2 | | | CA2 | | | MC2 | | |
|---|---|---|---|---|---|---|---|---|---|---|---|---|
| | Obj | Gap% | PI | Obj | Gap% | PI | Obj | Gap% | PI($\times 10^3$) | Obj | Gap% | PI |
| SCIP | 306.06 | 4.51 | 14953 | -1325.80 | 3.45 | 9542.1 | -185914 | 17.87 | 12312 | -1702 | 8.38 | 30039 |
| SCIP (500s) | 300.25 | 2.74 | / | -1361.33 | 0.86 | / | -207856 | 8.18 | / | -1704 | 8.26 | / |
| SCIP (1000s) | 296.18 | 1.37 | / | -1366.06 | 0.52 | / | -214754 | 5.13 | / | -1707 | 8.13 | / |
| U-LNS | 304.28 | 3.96 | 14268 | -1359.86 | 0.97 | 2778.5 | -207054 | 8.53 | 8032.5 | -1727 | 7.03 | 24862 |
| R-LNS | 304.24 | 3.94 | 14392 | -1363.30 | 0.71 | 2079.3 | -212024 | 6.34 | 7050.0 | -1737 | 6.52 | 22450 |
| FT-LNS | 306.10 | 4.49 | 14885 | -1359.90 | 0.96 | 2765.6 | -205812 | 9.08 | 8324.2 | -1738 | 6.44 | 22347 |
| DINS | 301.55 | 2.99 | 13916 | -1364.22 | 0.65 | 1935.4 | -212523 | 6.11 | 6848.5 | -1727 | 7.02 | 24815 |
| GINS | 302.33 | 3.14 | 14008 | -1363.17 | 0.69 | 2011.5 | -210539 | 6.52 | 7433.7 | -1737 | 6.52 | 22477 |
| RINS | 301.29 | 2.95 | 13793 | -1365.52 | 0.58 | 1844.7 | -211367 | 6.55 | 7129.3 | -1732 | 6.75 | 23619 |
| RENS | 300.42 | 2.78 | 13465 | -1365.71 | 0.55 | 1782.6 | -212789 | 6.02 | 6735.2 | -1742 | 6.23 | 20959 |
| RL-LNS | 297.85 | 1.66 | 13007 | -1367.12 | 0.51 | 1524.7 | -216255 | 4.13 | 5933.4 | -1803 | 3.20 | 8449.6 |
| Branching | 296.88 | 1.53 | 12916 | -1365.91 | 0.55 | 1769.4 | -215379 | 4.52 | 6142.7 | -1805 | 3.19 | 7857.3 |
| **BTBS-LNS-L** | **293.56** | **0.51** | **12431** | -1372.66 | 0.04 | 543.69 | **-222590** | **1.67** | **4800.3** | -1831 | 1.45 | 3385.9 |
| **BTBS-LNS-G** | 294.05 | 0.68 | 12498 | -1372.89 | 0.012 | 515.28 | -222075 | 1.89 | 5012.6 | -1831 | 1.44 | 3397.5 |
| Gurobi | 294.12 | 0.71 | 12528 | **-1373.14** | **0.01** | 495.88 | -218245 | 3.60 | 5723.5 | **-1839** | **1.01** | 2195.6 |
| Methods | SC4 | | | MIS4 | | | CA4 | | | MC4 | | |
| | Obj | Gap% | PI | Obj | Gap% | PI | Obj | Gap% | PI($\times 10^3$) | Obj | Gap% | PI |
| SCIP | 178.1 | 5.41 | 15524 | -2654.38 | 3.45 | 22745 | -371580 | 16.61 | 25275 | -3397 | 8.71 | 78510 |
| SCIP (500s) | 175.8 | 4.21 | / | -2654.45 | 3.44 | / | -371580 | 16.61 | / | -3398 | 8.69 | / |
| SCIP (1000s) | 173.8 | 3.05 | / | -2665.90 | 3.03 | / | -371580 | 16.61 | / | -3406 | 8.46 | / |
| U-LNS | 174.4 | 3.42 | 14814 | -2710.42 | 1.41 | 9759.0 | -412510 | 7.42 | 16470 | -3446 | 7.39 | 68245 |
| R-LNS | 174.0 | 3.26 | 14747 | -2722.10 | 0.98 | 7745.5 | -418014 | 6.19 | 15875 | -3461 | 6.98 | 64712 |
| FT-LNS | 175.0 | 3.75 | 14882 | -2713.50 | 1.30 | 9150.3 | -408611 | 8.30 | 17328 | -3459 | 7.02 | 65329 |
| DINS | 173.9 | 3.23 | 14725 | -2720.33 | 1.03 | 7982.4 | -420542 | 5.02 | 14789 | -3461 | 6.97 | 64593 |
| GINS | 174.1 | 3.28 | 14782 | -2725.41 | 0.85 | 7244.7 | -418751 | 5.99 | 15538 | -3459 | 7.04 | 65778 |
| RINS | 173.5 | 2.96 | 14599 | -2718.72 | 1.09 | 8218.0 | -419592 | 5.78 | 15309 | -3463 | 6.89 | 63575 |
| RENS | 173.4 | 2.95 | 14573 | -2727.11 | 0.82 | 6972.1 | -420311 | 5.17 | 14916 | -3464 | 6.85 | 62998 |
| RL-LNS | 175.2 | 3.73 | 14866 | -2735.41 | 0.57 | 5365.1 | -427433 | 3.52 | 13572 | -3587 | 3.76 | 39645 |
| Branching | 174.5 | 3.39 | 14689 | -2732.82 | 0.64 | 5744.8 | -428325 | 3.37 | 13349 | -3569 | 4.21 | 42718 |
| **BTBS-LNS-L** | **169.8** | **0.84** | **13716** | -2747.04 | 0.07 | 2140.4 | **-439431** | **1.39** | **11128** | -3664 | 1.52 | 21195 |
| **BTBS-LNS-G** | 170.5 | 1.20 | 13789 | -2745.15 | 0.11 | 2636.9 | -437522 | 1.46 | 11705 | **-3666** | **1.51** | 20984 |
| Gurobi | 170.5 | 1.22 | 13795 | **-2748.02** | **0.04** | 2215.7 | -389396 | 12.61 | 21959 | -3521 | 5.38 | 51298 |

**3) FT-LNS** (Song et al., 2020), **RL-LNS** (Wu et al., 2021a) and **Branching** (Sonnerat et al., 2021): some learning-based LNS policies.

**4) LNS-TG, LNS-Branch, LNS-IBT, LNS-IT, LNS-ATT, BTBS-LNS-F**: Degraded versions of **BTBS-LNS** to test the effectiveness of each component. Details can refer to the Appendix. A.3.

**Instances.** It covers both binary and MIP problems. We follow Wu et al. (2021a) to test our approach on four NP-hard binary Integer Programming Problems: Set Covering (SC), Maximal Independent Set (MIS), Combinatorial Auction (CA), and Maximum Cut (MC). We generate 200, 20, and 100 instances as training, validation, and testing sets, respectively. To evaluate the generalization ability, we also generate scale-transfer test instances, such as SC2, and MIS4 in Table 2. The suffix number refers to instance scales, for which the details are gathered in Table. 6 in Appendix A.3.

We also test our method on two NP-hard MIP datasets provided in Machine Learning for Combinatorial Optimization (ML4CO) competition[2]: Balanced Item Placement (**Item**) and Anonymous MIPLIB (**AMIPLIB**), on their official testing instances. Balanced Item Placement contained 1050

---
[2] https://www.ecole.ai/2021/ml4co-competition/

binary variables, 33 continuous variables, and 195 constraints per instance. The anonymous MIPLIB consists of a curated set of instances from MIPLIB, which is a long-standing standard benchmark for MIP solvers, with diverse problem distributions, in which general integer variables are included. We also show empirical results on the whole MIPLIB benchmark set in Appendix A.5 and per-instance comparison in Appendix B, where our **BTBS-LNS** even surpasses Gurobi on average.

**Hyperparameters.** We run experiments on Intel(R) Xeon(R) E5-2678 2.50GHz CPU. Performance comparison on CPU **vs** GPU version of our approach are discussed in Appendix A.9. Note that all the approaches were evaluated with three different seeds, and the average performance was reported (see detail stability analysis in Appendix A.8). We use the open source SCIP[3] (v7.0.3) as the baseline solver by default (recall the blue box in Fig. 1). Gurobi version experiments are gathered in Appendix. A.6. We train 20 epochs for each instance, with 50 iterations per epoch and 2s re-optimization time limit per iteration. LNS and branching are trained sequentially, with RL (see see Sec. 3.3) and imitation learning (see Sec. 3.4), respectively. The embedding for nodes and edges were both 64-dimensional vectors. Specifically for branching, we set the max branching variables $k = 50$ in Eq. 6 for local branching variant. In the inference phase, the branching variable ratio $r$ in Alg. 2 are empirically set to 10% for both branching variants. Note that **BTBS-LNS** by default denotes the local branching variant **BTBS-LNS-L** throughout this paper.

**Evaluation metric.** As the problems are too large to be solved in a reasonable time, we calculate the primal gap (Nair et al., 2020b) to measure the gap between the current solution $\mathbf{x}$ and the best-known solution $\mathbf{x}^*$ found by all methods, within a fixed time bound $T$:

$$gap = \frac{|\mathbf{c}^\top \mathbf{x} - \mathbf{c}^\top \mathbf{x}^*|}{\max(|\mathbf{c}^\top \mathbf{x}|, |\mathbf{c}^\top \mathbf{x}^*|)} \tag{7}$$

We also calculate Primal Integral (PI) to evaluate the anytime performance within the time limit:

$$PI = \int_{t=0}^{T} \mathbf{c}^\top \mathbf{x}_t dt - T\mathbf{c}^\top \mathbf{x}^* \tag{8}$$

where $\mathbf{x}_t$ denotes the best feasible solution within time $t$.

## 4.2 OVERALL PERFORMANCE EVALUATION

Table 1 compares the results on integer programming. As can be seen, compared with SCIP and all competing LNS baselines, both **BTBS-LNS-G** and **BTBS-LNS-L** achieves consistently superior performance across all problems. LNS-TG, LNS-Branch, and LNS-ATT are degraded versions of **BTBS-LNS** and they all perform slightly worse, revealing the effectiveness of attention-based tripartite graph and the extra branching policy. And comparing the two variants, **BTBS-LNS-G** delivers consistently superior performance over **BTBS-LNS-L**, and it even surpasses the leading commercial solver on SC, CA and MC. Note that detailed anytime performance on these instances are shown in Fig. 5 to Fig. 8 in Appendix A.4, further revealing the effectiveness of **BTBS-LNS**.

We also test our method on two NP-hard MIP problems, and the results are gathered in Table 3. Note that the anytime primal gap comparison are also shown in Fig. 3. Our method consistently outperforms SCIP and the competing LNS baselines, and is slightly worse than Gurobi, capable of finding even better solutions for around 27% test instances on both Item and AMIPLIB.

Specifically for the AMIPLIB problem, it contains a curated set of instances from MIPLIB. We split the instances into train, validation, and test sets by 70%, 15%, and 15% with cross-validation to test full performance. Policies learned from diverse training instances are directly applied to the test set. Note that we increase the solving and re-optimization time limit at each step to 1800s and 60s for both the training and testing phase, as they are too large to be solved. Different from Wu et al. (2021a), we

Table 3: Performance on MIP instances.

| Methods | Item | | | AMIPLIB |
| --- | --- | --- | --- | --- |
| | Obj | Gap% | PI | Gap% |
| SCIP | 23.33 | 50.73 | 4152.4 | 13.72 |
| SCIP (500s) | 19.83 | 39.41 | / | / |
| SCIP (1000s) | 17.02 | 31.05 | / | / |
| U-LNS | 20.39 | 44.29 | 3685.6 | 15.73 |
| R-LNS | 20.04 | 43.64 | 3485.0 | 14.96 |
| FT-LNS | 20.04 | 43.58 | 3498.5 | 12.55 |
| DINS | 18.08 | 37.23 | 3075.9 | 13.10 |
| GINS | 19.78 | 42.11 | 3514.7 | 13.64 |
| RINS | 20.53 | 44.88 | 3662.5 | 13.89 |
| RENS | 17.51 | 34.18 | 2925.0 | 11.75 |
| Branching | 18.84 | 40.12 | 3237.6 | 12.95 |
| LNS-TG | 18.05 | 37.85 | 3090.5 | 6.45 |
| LNS-Branch | 20.12 | 43.90 | 3537.0 | 9.32 |
| LNS-ATT | 15.54 | 26.91 | 2512.8 | 5.45 |
| LNS-IBT | / | / | / | 7.63 |
| LNS-IT | / | / | / | 7.65 |
| **BTBS-LNS-L** | 13.82 | 16.82 | 2030.3 | 4.19 |
| **BTBS-LNS-G** | 13.45 | 15.78 | 1912.5 | 4.35 |
| **BTBS-LNS-F** | / | / | / | 7.01 |
| Gurobi | **12.67** | **6.73** | **1895.6** | **0.81** |

---

[3] https://www.scipopt.org/

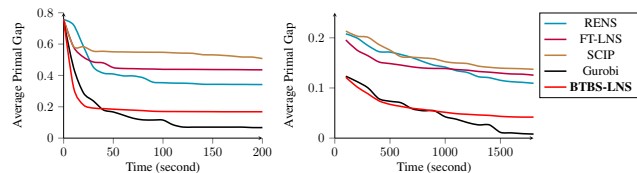

Figure 3: Performance on Balanced Item Placement (Left) & AMIPLIB (Right).

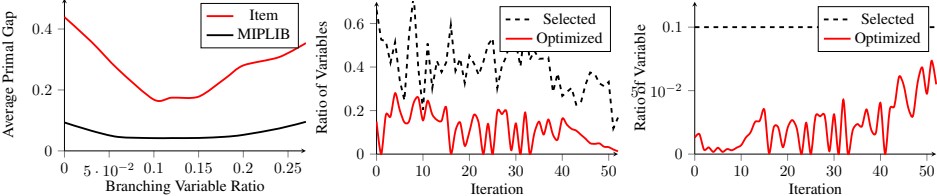

Figure 4: Impact of different branching ratios (Left). Selected & Optimized variables by LNS policy (Middle) & Branching Policy (Right) on Balanced Item Placement instances. *selected* means variables filtered by the learned branch & search policy; *optimized* denotes the updated variables.

consistently utilize open-source SCIP as the baseline solver. As seen from Table 3 and Fig. 3, our method significantly outperforms SCIP and LNS baselines, and even deliver slightly better performance than Gurobi at an early stage. LNS-IBT, LNS-IT and **BTBS-LNS-F** achieve significantly inferior performance than our **BTBS-LNS**, illustrating the effect of "Binarized Tightening" technique and its superior performance over Nair et al. (2020b).

### 4.3 PROBLEM-SCALE GENERALIZATION ABILITY STUDY

We test the generalization ability in line with Wu et al. (2021a) with 200s time limit. We directly use the trained policies on small-scale problems in Sec. 4.2. The results are gathered in Table 2.

As can be seen, the two variants show similar performance on the generalized instances. And compared with SCIP and all the competing LNS baselines, our approach still delivers significantly superior performance, showing a better generalization ability. As the problem sizes become larger, it can produce even better results than Gurobi on SC2, SC4, CA2, CA4, and MC4, and only slightly inferior on the remaining 3 groups. It suggests that our policies can be sometimes more efficient for larger instances than the leading commercial solver. Notably, there is a large gap between **BTBS-LNS** and Gurobi for Combinatorial Auction (CA), especially on CA4.

### 4.4 BRANCHING POLICY STUDY BY VARIABLE RATIOS

To enhance exploration, an extra branching policy was trained and utilized to help the pure LNS escape local optimum. Fig. 4 (**Left**) depicts the impact of branching variables ratios $r$ (see Alg. 2).

When the ratio is small ($< 0.1$), a larger branching size leads to a better performance. In fact, the leverage of branching can be regarded as a correction for LNS, facilitating it to escape local optimum. Fig. 4 (**Right**) depicts the selected and updated variable ratios. Branch and search policy adaptively select different variable subsets for re-optimization. However, when the branching size becoming extremely large, the performance significantly degrades limited by the solving ability.

## 5 CONCLUSION AND OUTLOOK

We have proposed a binarized tightening branch and search approach to learn LNS policies. It was designed to efficiently deal with general MIP problems, and delivers superior performance over numerous competing baselines, including MIP solvers, learning and heuristic based LNS approaches, on ILP, MIP datasets and even heterogeneous instances from MIPLIB. Sufficient ablation studies demonstrate the effectiveness of each component, including the tripartite graph, binarize and tighten scheme, and the extra branching at each step.

However, the proposed approach is only a primal heuristic that cannot prove optimality, which are also common limitations of LNS-based approaches. Implementing them into MIP solvers as primal heuristics may be a possible solution. However, interaction with current existed primal heuristics, and the rule to take effect, are key challenges in practical implementation. In general, applications of the learning-based approach in real-world scenarios will be our future directions.

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

# A APPENDIX

## A.1 FURTHER DISCUSSION ON RELATED WORK

Table 4: Comparison of our method to existing works.

| References | Applicability | Approach | Addressing Local Optima | Training |
|---|---|---|---|---|
| Huang et al. (2023a) | Binary | Branching Relaxation | Adaptive Neighborhood Size | / |
| Huang et al. (2023b) | Binary | LNS | Adaptive Neighborhood Size | Contrastive Learning |
| Liu et al. (2022) | Binary | Local Branching | RL-based Branching Size | Regression + RL |
| Nair et al. (2020a) | Binary | LNS | / | RL |
| Ding et al. (2020) | Binary | Solution Prediction | / | Imitation |
| Song et al. (2020) | Binary | LNS | / | Imitation & RL |
| Wu et al. (2021a) | Binary | LNS | / | RL |
| Hendel (2022) | General MIP | ALNS (Heuristic in B&B) | Adaptive Control for Mutiple Heuristics | Multi-armed Bandit |
| Paulus et al. (2022a) | General MIP | Learn to cut | / | Imitation |
| Sonnerat et al. (2021) | General MIP | LNS | Adaptive Neighborhood Size | Imitation |
| Nair et al. (2020b) | General MIP | Diving | / | Imitation (Generative Model) |
| **BTBS-LNS (Ours)** | General MIP | Branching on top of LNS | **Global (Local) Branching** | **RL (LNS) + Imitation (Branching)** |

**Table 4 compares our approach with some existing works in detail**. The core contribution of our **BTBS-LNS** is the general applicability and the addressing for local optima. As can be seen, most LNS-based papers (Liu et al., 2022; Nair et al., 2020a; Ding et al., 2020; Song et al., 2020; Wu et al., 2021a) solely deal with the binary programming problems due to its simpleness. Recently, some studies try to address the general MIP problems (Hendel, 2022; Sonnerat et al., 2021; Paulus et al., 2022a), in which Nair et al. (2020b) proposed a similar "bound tightening" technique. They differ with our approach in the following aspects. On one hand, the binary decision for each encoded variable was only applied for bound tightening, rather than directly fixed similar to Nair et al. (2020b). And on the other hand, current solution value was also considered in bound tightening decisions in our approach. Variables that sit far from both bounds may have a significantly wider exploration scope than close-to-bound variables, as they showed no explicit "preference" on either direction. In addition, our approach can easily transfer to unbounded variables as illustrated in Alg. 1. We made detailed comparison between the two approaches in Table. 3 and Table. 10. As can be seen, our **BTBS-LNS** consistently outperforms **BTBS-LNS-F**, demonstrating the effectiveness of our novel "Binarized Tightening" technique.

As for the local optima challenge, a few studies have tried Adaptive Neighborhood Size (ANS) (Huang et al., 2023a;b; Sonnerat et al., 2021) or hybrid heuristics control (Hendel, 2022), while it still requires hand-crafted hyperparameters, which are essential but difficult to determine. To address it more adaptively, we proposed to combine branching on top of pure LNS. When trapped in local optima, branching mechanism has the potential to select those wrongly-fixed backdoor variables by pure LNS for re-optimization. It is important to note that the concept of branching extends beyond the confines of the local branching (Sonnerat et al., 2021) and we also devised a novel variant termed "global branching" (see Sec. 3.4), which can deliver even better performance in some cases. In addition, the major difference between our hybrid framework and the pure local branching approach (Sonnerat et al., 2021) lies in that we concentrates solely on variables fixed by LNS to correct its decisions, rather than the whole variable set. This specificity arises from the observation that LNS frequently converges to local optima when a limited number of backdoor variables are inaccurately fixed. Empirical results in Table. 1, 2 and 3 demonstrated that our **BTBS-LNS** consistently outperforms the Branching baseline by Sonnerat et al. (2021).

We further review other studies related to ours, which can be divided into two categories: One is learning-based methods for specific MIP problems and the other is for general MIP problems.

### A.1.1 POLICY LEARNING FOR SPECIFIC MIP PROBLEMS

MIP problems cover numerous real-world tasks in many fields (Paschos, 2014) and quite a few studies attempt to solve certain types of problems, such as Traveling Salesman Problem (TSP) and Vehicle Routing Problems (VRP) (Li et al., 2021; Lu et al., 2019a), etc. The algorithms can be divided into two folds, construction methods and learned improvement heuristics.

Construction methods usually attempt to directly learn approximate optimal solutions. For example, different models, like Graph Pointer Networks (GPNs) (Ma et al., 2019) and Monte Carlo tree search (Xing & Tu, 2020; Fu et al., 2021) were both proposed to solve TSP instances, and Zhang et al. (2020) trains a policy to learn priority dispatching rules for scheduling problems via an end-to-end deep reinforcement learning agent.

Compared to construction models, methods that learn improvement heuristics can often deliver better performance, by learning to iteratively improve the solution (Wu et al., 2021a). The improvement heuristics can be a guide for next solution selection (Wu et al., 2021b), or policy to pick heuristics (Chen & Tian, 2019), or refinement from current solution (Lu et al., 2019b; Li et al., 2020), which all have demonstrated the effectiveness in routing and scheduling problems. In general, both the learned improvement heuristics and construction methods have proved validity in some specific problems. In contrast, this paper aims to solve general MIP problems by learning improvement heuristic policies.

### A.1.2 Learning to solve general MIP problems

Dual and primal are two main perspectives to improve solving efficiency for general MIP problems. Specifically, dual view aims to improve inner policies of Branch and Bound (B&B), e.g., variable selection (Gasse et al., 2019; Zarpellon et al., 2021; Gupta et al., 2020), node selection (He et al., 2014) and cut selection (Tang et al., 2020; Paulus et al., 2022b;a). With a better decision at each node, the overall solving process can be greatly simplified.

Different from the dual view, in the primal perspective, the algorithms aim to find better feasible solutions by prediction or learning-based heuristics. For example, Ding et al. (2020) learned a tripartite graph based deep neural network to generate partial assignments for binary variables, and in order to deal with the general integer variables, Nair et al. (2020b) proposed a bound tightening mechanism and learned partial assignments for each bit, respectively. Nevertheless, they were only applied in neural diving, and directly fixing may also lead to performance degradation, or even infeasible. To obtain broader applicability, learning-based primal heuristics, like large neighborhood search (Huang et al., 2023b; Song et al., 2020; Sonnerat et al., 2021; Nair et al., 2020a), local branching (Liu et al., 2022), gradually catch the attention of researchers.

In this paper, we mainly focus on large neighborhood search heuristics, which have achieved remarkable progress in recent years. For example, Hendel (2022) designed an adaptive approach to combine multiple existed LNS heuristics to enhance performance of single policy, while it is largely limited by the rule-based heuristics and requires hand-crafted hyperparameters. To make it further, learning a better neighborhood function have been more and more popular in recent years. Sonnerat et al. (2021) utilized imitation learning to select variable subsets to optimize at each step. Similarly, Song et al. (2020) also proposed a decomposition-based framework with imitation learning to learn the best variable decomposition. However, the imitation learning framework and the equal-size subsets makes it inflexible and dramatically limit the performance of learned policies. In this respect, Wu et al. (2021a) factorize the LNS policy into elementary actions on each variable, and trained a RL-based policy to select variable subsets dynamically. However, they cannot generalize to general integer variables and the local search drawbacks make it easy to converge in local optimum.

In general, current studies on LNS mainly focus on binary variables, and local search properties interfere with the performance in some complicated scenarios. In this respect, we propose a binarized-tightening branch and search approach to learn more efficient LNS policies for general problems.

### A.2 Hybrid Branch and Search

In this paper, we proposed a hybrid binarized tightening branch and search framework for general MIP problems. We tend to illustrate some details about the framework. Specifically, Alg. 1 depicts the pipeline of bound-tightening technique for each general integer variable, where we represent them with $d$ substitute binary variables and tighten the original variable bounds w.r.t the current solution value $p$ and bit-wise LNS decision $a_{i,j}^t$.

Alg. 3 depicts the overall training pipeline for the offline graph based branching policy. Specifically, we make the branching policy into a binary decision process (branch or not) for each variable, and utilize the cross-entropy loss to train the graph-based branching network. The output probability can help filter the potentially wrongly-fixed backdoor variables in a global or local perspective. The

---

**Algorithm 3** Offline training of branching policy for LNS

---

**Input**: tripartite graph based states $S = \{s_t | t = 1, 2, ..., n\}$
LNS decisions at each step $N = \{n_t | t = 1, 2, ..., n\}$
branching variable labels $B = \{b_t | t = 1, 2, ..., n\}$ collected from the local or global branching;
**Output**: trained policy $\pi_\theta(B | S, N)$

 1: *// Samples are collected by resolving the training instances, along with the learned LNS;*
 2: Let $D = \{((s_t, n_t), b_t) | t = 1, 2, ..., n\}$.
 3: *// train the model;*
 4: Initialize all learnable parameters $\theta$;
 5: **while** stopping criteria not meet **do**
 6:      Randomly select a batch of instances $D_C$ from D;
 7:      Optimize $\theta$ by minimizing cross-entropy loss;
 8: **end while**

---

Table 5: Training, Validation and Test accuracy for graph based branching network.

| | Local Branching | | | | | | Global Branching | | | | | |
|---|---|---|---|---|---|---|---|---|---|---|---|---|
| | SC | MIS | CA | MC | Item | AMIPLIB | SC | MIS | CA | MC | Item | AMIPLIB |
| Train% | 89.5 | 84.9 | 79.6 | 86.3 | 85.5 | 77.5 | 86.9 | 87.3 | 81.5 | 88.5 | 83.4 | 75.9 |
| Validation% | 84.8 | 83.5 | 75.1 | 82.1 | 82.8 | 74.9 | 83.7 | 84.9 | 80.9 | 87.0 | 81.8 | 75.1 |
| Test% | 82.5 | 81.6 | 72.9 | 80.5 | 81.5 | 74.2 | 83.1 | 82.6 | 80.1 | 84.5 | 80.7 | 73.8 |

Table 6: Average variable/constraints of instances

| Num of | Training | | | | Generalization | | | | | | | |
|---|---|---|---|---|---|---|---|---|---|---|---|---|
| | SC | MIS | CA | MC | SC2 | MIS2 | CA2 | MC2 | SC4 | MIS4 | CA4 | MC4 |
| Variables | 1000 | 1500 | 4000 | 2975 | 2000 | 3000 | 8000 | 5975 | 4000 | 6000 | 16000 | 11975 |
| Constraints | 5000 | 5939 | 2674 | 4950 | 5000 | 11933 | 5344 | 9950 | 5000 | 23905 | 10717 | 19950 |

overall training, validation and testing accuracy on different problems are listed in Table 5, including both the local and global branching variants.

The hybrid branch and search framework works as in Alg. 2 in the main text, where we place the branching network on top of LNS. Specifically, $n_i^t$ denotes the LNS decision for each variable. Note that as illustrated above, there will be $d$ separate decisions for general integer variables, denoted as $n_{i,j}^t$. $b_i^t$ denotes the branching decision (branch or not) for each variable. The hybrid branch and search policy work together to formulate the sub-MIP at each step. It consists of three main steps, bound tightening in Line 3-4 for general integer variables, directly fixing in Line 6-7 for binary variables and extra branching in Line 9-10. Branching policy can be regarded as an approach to enhance the learned LNS policy by selecting and optimizing some wrongly-fixed variables by LNS (see Line 15-16).

### A.3    DETAIL FOR THE EXPERIMENTS

As discussed, the tripartite graph is utilized to represent the problem states in both the RL-based LNS policy and the offline branching policy. We describe in Table 7 the variable, constraint, objective, and multi-source edge features in detail. Except for the dynamic solving status, all the other features are collected at the root node of the B&B search tree, and the dynamic features are collected along with the optimization process.

The average variable and constraint size used in our experiments are listed in Table 6, which consists of small-scale training instances and some hard instances used for evaluating the generalization ability. And as illustrated in Sec. 4, we compare our proposed **BTBS-LNS** with various baselines, which are explained as follows in detail:

- **SCIP (v7.0.3)**: state-of-the-art open source solver with default settings. Note that SCIP is allowed to run for a longer time, i.e., 500s and 1000s.

- **Gurobi (v9.5.0)**: state-of-the-art commercial solver.

Table 7: Description of the tripartite graph features.

| Tensor | Feature Description |
|---|---|
| $\mathcal{V}$ | variable type (binary, integer, continuous). |
| | objective coefficient. |
| | lower and upper bound. |
| | reduced cost. |
| | solution value fractionality. |
| | **(dynamic)** solution value in incumbent. |
| | **(dynamic)** average solution value. |
| | **(dynamic)** best solution value. |
| | **(Branching Only)** LNS decisions at current step. |
| $\mathcal{C}$ | cosine similarity with objective. |
| | tightness indicator in LP solution. |
| | dual solution value. |
| | bias value, normalized with constraint coefficients |
| $\mathcal{O}$ | average states of related variables. |
| $\mathcal{V}$ - $\mathcal{C}$ | constraint coefficient per variable. |
| $\mathcal{V}$ - $\mathcal{O}$ | objective coefficient per variable. |
| $\mathcal{C}$ - $\mathcal{O}$ | constraint right-hand-side (RHS) coefficients. |

- **U-LNS**: an LNS version that uniformly samples variables at a fixed subset size. Note that for U-LNS, R-LNS and FT-LNS, we perform the same settings as those in Wu et al. (2021a).

- **R-LNS**: an LNS version (Song et al., 2020) that randomly groups variables into equal subsets and reoptimizes them.

- **DINS** (Ghosh, 2007), **GINS** (Maher et al., 2017), **RINS** (Danna et al., 2005) and **RENS** (Berthold, 2014): heuristic-based LNS policies that have been implemented in SCIP.

- **FT-LNS**: an LNS version (Song et al., 2020) that applies imitation learning to learn the best R-LNS policies.

- **RL-LNS**: A similar reinforcement learning LNS approach for variable subset optimization (Wu et al., 2021a), while mainly focused on binary variable optimization.

- **Branching** (Sonnerat et al., 2021): An LNS framework by imitation learning from the labels collected by incorporating local branching constraints.

- **LNS-TG**: A variant of our method, where we replace the tripartite graph with the widely used bipartite graph.

- **LNS-Branch**: A variant of our method, where we remove the branching policy.

- **LNS-IBT**: A variant of our method, where the general integer variables are equally treated as binary variables.

- **LNS-IT**: A variant of our method, where we remove the "Tightening" technique and fix the integer variable to its current solution when either bit is fixed.

- **LNS-ATT**: A variant of our method, where we replace our attention-based graph attention network with the widely used GNN.

- **BTBS-LNS-F**: A variant of our **BTBS-LNS**, where we replace our bound tightening mechanism with that proposed by Nair et al. (2020b).

Note that the work by Sonnerat et al. (2021) doesn't have open-source code and some hyperparameters are difficult to fine-tune in different MIP problems. However, in order to further evaluate our proposed framework with pure local branching based methods, we try to reproduce them. Some reproduction details are as follows:

**1)** For fair comparison, we replace the neural diving in Sonnerat et al. (2021) with an initial feasible solution generated by SCIP, the same as our approach.

**2)** In data collection, the desired Hamming radius $\eta_t$ are selected as 50, the same as our branching policy.

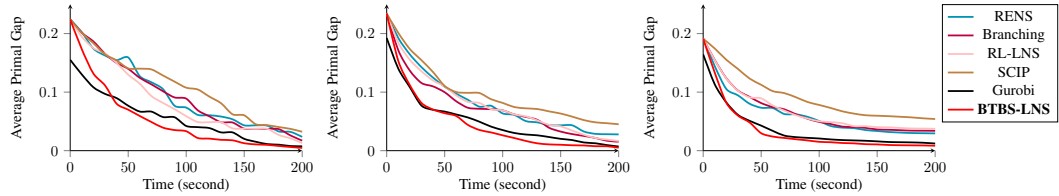

Figure 5: Anytime Performance on Set Covering (SC) problem and its scale-transfer instances. **From left to right**: Performance comparison on instances from SC, SC2, SC4. (see Table 6 for detail).

**3)** The model structure were the same as its descriptions, where we use the code provided by Gasse et al. (2019), and additionally use a fixed-size window (3 in the paper) of past variable assignments as variable features.

**4)** Loss function and training were all respect to the settings in the paper.

**5)** In the inference phase with the learned policy, we performed the same action sampling mechanism as in Sonnerat et al. (2021). As for the adaptive neighborhood size, we start with 10% of the integer variable size, and the dynamic factor $a$ was tuned from 1.01 to 1.05. Best-performing parameters will be selected for comparison in each problem. As a result, on SC and MIS, $a$ was set as 1.02, and $a = 1.03$ can deliver the best performance on other problems.

**6)** Reproduction details will also be public along with the code and data.

In addition, we do not make comparisons with Hendel (2022), as it is embedded in SCIP as a heuristic and difficult for fair comparison, and thus we solely tested the implemented heuristics separately, e.g.,RINS, DINS, RENS.

To further evaluate our approach on generalized ILP instances, we further increase the time limit to 500s and 1000s respectively on CA, with results shown in Table 8. Our method consistently outperforms Gurobi with the same time limit. For CA4, it can even produce better solutions with a much shorter time limit. It empirically requires over 3 hours for Gurobi to deliver the same primal gap on CA4, being $58\times$ slower than our method.

Table 8: Evaluation on CA against Gurobi.

| Methods | CA2 | | CA4 | |
|---|---|---|---|---|
| | Obj | Gap% | Obj | Gap% |
| Gurobi | -218245 | 3.60 | -389396 | 12.61 |
| Gurobi(500s) | -224245 | 0.95 | -431626 | 3.14 |
| Gurobi(1000s) | **-225629** | **0.33** | -436188 | 2.11 |
| **BTBS-LNS** | -222590 | 1.67 | -439431 | 1.39 |
| **BTBS-LNS**(500s) | -225108 | 0.56 | **-445563** | **0** |

Table 9: Evaluation by Gurobi as baseline solver.

| Methods | Item | | | AMIPLIB |
|---|---|---|---|---|
| | Obj | Gap% | PI | Gap% |
| U-LNS | 17.64 | 36.08 | 3004.3 | 6.44 |
| R-LNS | 16.62 | 31.94 | 2788.6 | 6.01 |
| FT-LNS | 15.64 | 27.31 | 2519.4 | 5.45 |
| **BTBS-LNS** | **12.27** | **4.56** | **1823.7** | **0.47** |
| Gurobi | 12.67 | 6.73 | 1895.6 | 0.81 |

### A.4 ANYTIME PERFORMANCE ON BINARY INTEGER PROGRAMMING PROBLEMS

In order to further evaluate the anytime performance among the competing approaches, we plot the anytime primal gap curves on four binary integer programming problems, Set Covering (SC), Maximal Independent Set (MIS), Combinatorial Auction (CA) and Maximum Cut (MC), respectively. The results are gathered in Figure 5, 6, 7, 8, respectively.

As seen from the results, our **BTBS-LNS** delivers consistently superior performance over the competing LNS baselines almost at any point, demonstrating its efficiency and effectiveness. More surprisingly, the proposed approach can achieve superior performance over the leading commercial solver in some cases, especially on the scale-transfer instances, purely by the learned policy on small-scale instances.

### A.5 SUPPLEMENTARY EXPERIMENTS ON MIPLIB

As illustrated in Sec. 4, we have tested the effectiveness of the proposed **BTBS-LNS** on Anonymous MIPLIB(**AMIPLIB**) from ML4CO 2021 competition[4]. To make further evaluation, especially on

---

[4]https://www.ecole.ai/2021/ml4co-competition/

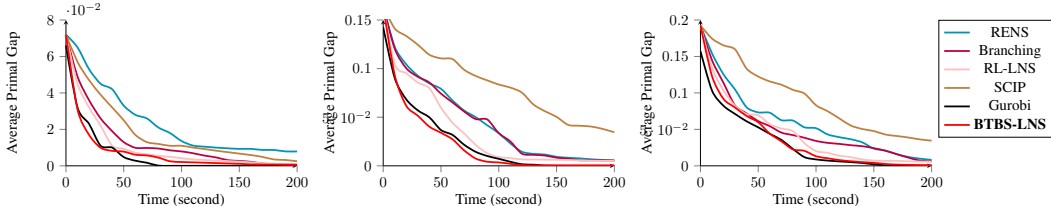

Figure 6: Anytime Performance on Maximal Independent Set (MIS) problem and its scale-transfer instances. **From left to right**: Performance comparison on instances from MIS, MIS2, MIS4. (see Table 6 for detail).

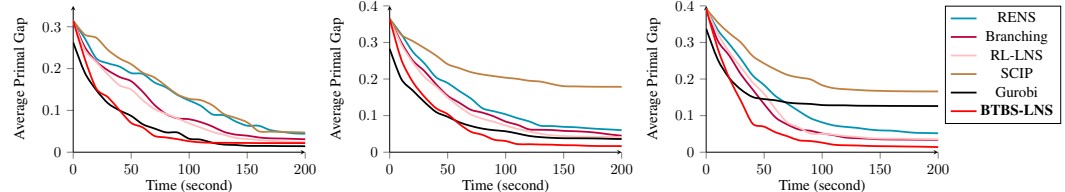

Figure 7: Anytime Performance on Combinatorial Auction (CA) problem and its scale-transfer instances. **From left to right**: Performance comparison on instances from CA, CA2, CA4. (see Table 6 for detail).

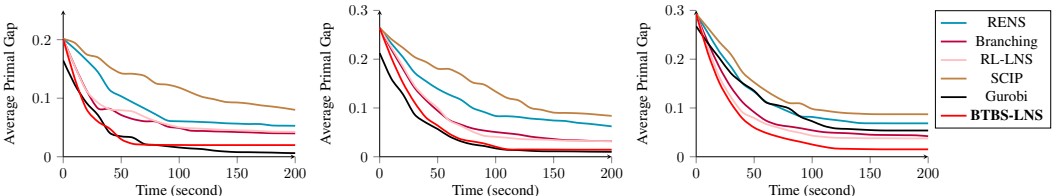

Figure 8: Anytime Performance on Maximum Cut (MC) problem and its scale-transfer instances. **From left to right**: Performance comparison on instances from MC, MC2, MC4. (see Table 6 for detail).

Table 10: Performance comparison on the whole MIPLIB benchmark set.

|  | SCIP | SCIP(600s) | SCIP(900s) | U-LNS | R-LNS | FT-LNS | **BTBS-LNS** | **BTBS-LNS**-F | Gurobi |
|---|---|---|---|---|---|---|---|---|---|
| Gap% | 15.15 | 11.08 | 8.79 | 16.26 | 15.94 | 13.07 | **1.75** | 3.11 | 1.98 |

some heterogeneous and hard instances, we also performed the experiments on the whole MIPLIB benchmark set[5], which is a standard test set used to compare the performance of mixed integer optimizers. The benchmark set contains 240 instances. We compared different methods in a 300s time limit, which is the geometric mean of solving time of the solved instances with SCIP, and the re-optimization time for each step was set as 5s. Other hyperparameters remain the same as **AMIPLIB** in Sec. 4. We perform **cross-validation** to make fair comparison and split them into training, validation, and testing sets by 70%, 15%, and 15% at each round. Policies learned from diverse training instances are directly applied to the test set.

The overall comparison results were gathered in Table 10. As can be seen, our **BTBS-LNS** can deliver significantly better results compared with all the competing baselines, including the leading commercial solver, indicating its effectiveness and generalization ability. Furthermore, we notice that **BTBS-LNS-F** performs slightly inferior than Gurobi and our approach, further revealing the superior performance of our Binarized Tightening technique over that proposed by Nair et al. (2020b). Detailed per-instance comparison are gathered in Table 17 in Appendix B.

In addition, as illustrated in Alg. 1, we devised a novel virtual bound technique specifically for unbounded integer variables. To evaluate its performance, we conducted an extensive analysis across all instances featured in MIPLIB benchmark set. Notably, there are 19 and 4 instances that contained unbounded integer variables before and after the presolve, respectively. In this section, we

---
[5]https://miplib.zib.de/

Table 11: Performance comparison on MIPLIB instances that contained unbounded variables.

| Instance | SCIP | U-LNS | R-LNS | FT-LNS | BTBS-LNSw/o ubd | BTBS-LNS | Gurobi |
|---|---|---|---|---|---|---|---|
| gen-ip054 | 6858.879 | 6858.879 | 6852.733 | 6858.879 | 6852.733 | 6852.733 | 6840.966* |
| gen-ip002 | -4783.733* | -4772.597 | -4772.597 | -4768.253 | -4783.733* | -4783.733* | -4783.733* |
| **neos-3046615-murg** | 1610 | 1670 | 1651 | 1651 | 1610 | 1607 | 1600* |
| **buildingenergy** | 42652.34 | 42652.34 | 42652.34 | 42652.34 | 34243.89 | 33324.73 | 33283.85* |

Table 12: Experiments with Gurobi as the baseline for binary Integer Programming (IP).

| Methods | SC | | | MIS | | | CA | | | MC | | |
|---|---|---|---|---|---|---|---|---|---|---|---|---|
| | Obj | Gap% | PI | Obj | Gap% | PI | Obj | Gap% | PI($\times 10^3$) | Obj | Gap% | PI |
| U-LNS | 559.74 | 2.59 | 18820 | -683.45 | 0.41 | 635.32 | -111036 | 3.78 | 2690.5 | -889.62 | 4.07 | 5633.8 |
| R-LNS | 560.49 | 3.01 | 18925 | -683.94 | 0.34 | 545.71 | -109797 | 4.85 | 2999.0 | -891.95 | 3.75 | 5189.5 |
| FT-LNS | 564.76 | 3.38 | 19521 | -684.16 | 0.73 | 462.45 | -110319 | 4.40 | 2856.4 | -891.06 | 3.79 | 5214.7 |
| RL-LNS | 549.16 | 1.57 | 16911 | -686.12 | 0.09 | 179.94 | -113862 | 1.37 | 2029.1 | -894.51 | 3.52 | 4812.5 |
| **BTBS-LNS** | **546.84** | **0.28** | **15987** | **-686.24** | **0** | **165.24** | **-115083** | **0.27** | **1710.6** | **-923.96** | **0.38** | **426.89** |
| Gurobi | 549.44 | 0.75 | 16796 | **-686.24** | **0** | 173.15 | -113731 | 1.44 | 2075.4 | -921.90 | 0.62 | 842 |

Table 13: Generalization to large-scale binary integer programming (IP) instances with Gurobi as the baseline.

| Methods | SC2 | | | MIS2 | | | CA2 | | | MC2 | | |
|---|---|---|---|---|---|---|---|---|---|---|---|---|
| | Obj | Gap% | PI | Obj | Gap% | PI | Obj | Gap% | PI($\times 10^3$) | Obj | Gap% | PI |
| U-LNS | 298.28 | 2.48 | 13599 | -1368.81 | 0.32 | 1551.6 | -219447 | 3.06 | 5442.1 | -1788 | 3.76 | 13713 |
| R-LNS | 300.17 | 2.73 | 14052 | -1365.68 | 0.55 | 1845.2 | -220497 | 2.60 | 5112.0 | -1789 | 3.75 | 13359 |
| FT-LNS | 302.08 | 3.29 | 14338 | -1369.30 | 0.28 | 1485.2 | -217950 | 3.72 | 5823.5 | -1783 | 4.04 | 13753 |
| **BTBS-LNS** | **292.88** | **0.28** | **12275** | **-1373.18** | **0** | **462.38** | **-225319** | **0.47** | **4125.1** | **-1858** | **0.01** | **350.45** |
| Gurobi | 294.12 | 0.71 | 12528 | -1373.14 | 0.01 | 495.88 | -218245 | 3.60 | 5723.5 | -1839 | 1.01 | 2195.6 |
| Methods | SC4 | | | MIS4 | | | CA4 | | | MC4 | | |
| | Obj | Gap% | PI | Obj | Gap% | PI | Obj | Gap% | PI($\times 10^3$) | Obj | Gap% | PI |
| U-LNS | 172.6 | 2.56 | 14150 | -2731.62 | 0.64 | 5515.7 | -427694 | 4.02 | 15712 | -3537 | 4.95 | 46965 |
| R-LNS | 171.9 | 2.36 | 14112 | -2734.72 | 0.52 | 4846.3 | -427992 | 3.95 | 15275 | -3541 | 4.84 | 46380 |
| FT-LNS | 174.2 | 3.34 | 14515 | -2734.22 | 0.54 | 4915.0 | -429190 | 3.68 | 14588 | -3543 | 4.78 | 45795 |
| **BTBS-LNS** | **168.8** | **0.27** | **13424** | **-2748.84** | **0.01** | **2051.8** | **-442616** | **0.67** | **10025** | **-3721** | **0** | **11034** |
| Gurobi | 170.5 | 1.22 | 13795 | -2748.02 | 0.04 | 2215.7 | -389396 | 12.61 | 21959 | -3521 | 5.38 | 51298 |

compared our **BTBS-LNS** with a variant **BTBS-LNSw/o ubd**, where the special handling for un-bounded integer variables (see Line 2-6 in Alg. 1) are removed. In other words, unbounded variables were free to optimize at each step. The comparison results on the four instances that still contain unbounded variables after presolve are gathered in Table. 11. As can be seen, our proposed **BTBS-LNS**, outperforms the variant **BTBS-LNSw/o ubd** on two instances and achieves parity on the other two. These findings underscore the potent effectiveness of our proposed bound tightening technique, substantiating its value in enhancing solution quality and optimization efficiency. We will continue the experimentation on more unbounded MIP problems in the future.

## A.6 SUPPLEMENTARY EXPERIMENTS WITH GUROBI

In order to evaluate the performance of different approaches with Gurobi as the baseline solver, we perform extensive experiments on MIP problems, four binary integer programming problems and their scale-transfer instances.

The hyperparameters remain unchanged as those in SCIP counterparts. The results on four binary integer programming problems and their scale-transfer instances are gathered in Table 12 and Table 13. And the comparison results on MIP problems are reported in Table 9. As can be seen, our **BTBS-LNS** consistently outperforms Gurobi across all the problems with different sizes, indicating the effectiveness and generalization ability to different solvers.

## A.7 EVALUATION ON OUR PROPOSED ATTENTION APPROACH

As illustrated in Sec. 3.3, we proposed a slightly different attention approach for the Graph Attention Network (Veličković et al., 2018), where we remove the commonly-utilized Softmax-normalized formulation. Specifically, for a node $i$ in the tripartite graph, the weight coefficient $w_{ij}$ across all neighboring nodes $j \in N_i$ are simply averaging by $|N_i|$ (see Eq. 4) to fully reserve the absolute importance between nodes, rather than Softmax normalized in the general handling.

Table 14: Performance comparison for different attention approaches on four binary Integer Programming problems: SC, MIS, CA, MC.

| Methods | SC | | | MIS | | | CA | | | MC | | |
|---|---|---|---|---|---|---|---|---|---|---|---|---|
| | Obj | Gap% | PI | Obj | Gap% | PI | Obj | Gap% | PI ($\times 10^3$) | Obj | Gap% | PI |
| LNS-Softmax | 548.22 | 0.56 | 16493 | -685.82 | 0.05 | 181.75 | -112810 | 2.27 | 2233.6 | -906.15 | 1.98 | 1755.0 |
| **BTBS-LNS** | **547.88** | **0.47** | **16234** | **-685.86** | **0.05** | **181.47** | **-112864** | **2.18** | **2196.8** | **-909.17** | **1.99** | **2518** |

Table 15: Average Standard Deviations for our proposed **BTBS-LNS** on different problems.

| Methods | SC | | SC2 | | SC4 | |
|---|---|---|---|---|---|---|
| | Obj | Gap% | Obj | Gap% | Obj | Gap% |
| **BTBS-LNS** | $547.88 \pm 0.59\%$ | $0.47 \pm 0.88\%$ | $293.56 \pm 0.77\%$ | $0.51 \pm 0.68\%$ | $169.80 \pm 0.68\%$ | $0.84 \pm 1.01\%$ |
| Methods | MIS | | MIS2 | | MIS4 | |
| | Obj | Gap% | Obj | Gap% | Obj | Gap% |
| **BTBS-LNS** | $-685.86 \pm 0.74\%$ | $0.05 \pm 0.78\%$ | $-1372.66 \pm 0.51\%$ | $0.04 \pm 0.21\%$ | $-2747.04 \pm 0.32\%$ | $0.07 \pm 0.19\%$ |
| Methods | CA | | CA2 | | CA4 | |
| | Obj | Gap% | Obj | Gap% | Obj | Gap% |
| **BTBS-LNS** | $-112864 \pm 0.32\%$ | $2.18 \pm 0.29\%$ | $-222590 \pm 0.39\%$ | $1.67 \pm 0.41\%$ | $-439431 \pm 0.33\%$ | $1.39 \pm 0.49\%$ |
| Methods | MC | | MC2 | | MC4 | |
| | Obj | Gap% | Obj | Gap% | Obj | Gap% |
| **BTBS-LNS** | $-909.17 \pm 0.48\%$ | $1.99 \pm 0.52\%$ | $-1831.00 \pm 0.66\%$ | $1.45 \pm 0.58\%$ | $-3664 \pm 0.73\%$ | $1.52 \pm 0.84\%$ |
| Methods | Item | | AMIPLIB | | MIPLIB | |
| | Obj | Gap% | Obj | Gap% | Obj | Gap% |
| **BTBS-LNS** | $13.82 \pm 1.09\%$ | $16.82 \pm 0.96\%$ | / | $4.19 \pm 1.51\%$ | / | $1.75 \pm 1.62\%$ |

To further evaluate the performance of different attention approaches, we compare our proposed **BTBS-LNS** with **LNS-Softmax**, where we instead utilized the common Softmax normalized approach and all the others remain the same. We performed the comparison on four binary programming problems, and the results are gathered in Table 14. As can be seen, with our updated attention mechanism, **BTBS-LNS** can obtain consistently superior performance over **LNS-Softmax**, revealing the effectiveness of our novel attention approach.

## A.8 STABILITY ANALYSIS OF OUR APPROACH

As illustrated in Sec. 4, all the experiments were performed with three different seeds to make fair comparison for different approaches. The average standard deviations for our proposed **BTBS-LNS** on different problems are gathered in Table. 15. As can be seen, the **BTBS-LNS** is fairly robust to different seeds, with average standard deviations lower than 2% even on some heterogeneous instances, like MIPLIB.

## A.9 EXPERIMENTS WITH CPU VS GPU

As illustrated in Sec. 4, all the experiments were performed on the Intel(R) Xeon(R) E5-2678 v3 2.50GHz CPU with 4 physical cores, and it achieved competitive performance compared even with the leading commercial solver. In this section, we will further test the GPU version (NVIDIA GeForce RTX 2080) of our proposed **BTBS-LNS** on the balanced item placement problem, and the results are given in Table 16.

Fig. 9 further depicts the anytime primal gap comparison between CPU and GPU version in detail. As can be seen from the results, compared with CPU implementation, GPU version **BTBS-LNS** delivers slightly better performance, in which the overall primal gap and primal integral improve by 0.83% and 0.99%, respectively. In other words, our proposed **BTBS-LNS** may achieve even better performance when implemented in GPU environment.

Table 16: Performance Analysis (GPU **vs** CPU).

| Methods | Item | | |
|---|---|---|---|
| | Obj | Gap% | PI |
| **BTBS-LNS** + CPU | 13.82 | 16.82 | 2030.3 |
| **BTBS-LNS** + GPU | 13.78 | 16.68 | 2010.2 |

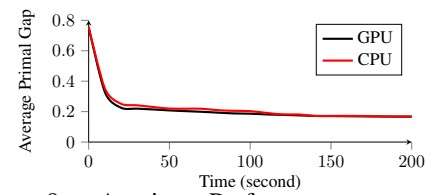

Figure 9: Anytime Performance comparison (GPU **vs** CPU).

# B  PER-INSTANCE PERFORMANCE COMPARISON ON MIPLIB

Considering that the results on MIPLIB instances may deliver high variances due to the significantly different problem distributions across instances, showing only the average gap like Table 10 may be not sufficient. In this respect, we report the detail per-instance performance within the given time limit on the competing approaches, and the results are gathered in Table 17.

Note that we only report **218**/240 instances from MIPLIB benchmark set. The following instances were removed, as no feasible solution can be found for them within the pre-defined time limit:

**1) Instances that are infeasible (6)**:

- bnatt500
- cryptanalysiskb128n5obj14
- fhnw-binpack4-4
- neos-2075418-temuka
- neos-3988577-wolgan
- neos859080

**2) Instances that cannot generate feasible solution by the baseline solver within timelimit (16)**:

- cryptanalysiskb128n5obj16
- gfd-schedulen180f7d50m30k18
- highschool1-aigio
- irish-electricity
- neos-1354092
- neos-3402454-bohle
- neos-4532248-waihi
- neos-5104907-jarama
- neos-5114902-kasavu
- ns1116954
- ns1952667
- peg-solitaire-a3
- physiciansched3-3
- rail02
- supportcase19
- supportcase22

Table 17: Per-instance performance comparison on MIPLIB 2017

| Instance | SCIP | SCIP(600s) | SCIP(900s) | U-LNS | R-LNS | FT-LNS | BTBS-LNS | BTBS-LNS-F | Gurobi | Best Solution |
|---|---|---|---|---|---|---|---|---|---|---|
| 30n20b8 | 302 | 302 | 302 | 353 | 302 | 302 | 302 | 302 | 302 | 302 |
| 50v-10 | 3340.37 | 3316.92 | 3313.18 | 3324.38 | 3340.37 | 3334.01 | 3311.18 | 3315.24 | 3311.18 | 3311.18 |
| academictimetablesmall | 228 | 228 | 228 | 228 | 228 | 228 | 45 | 45 | 6 | 0 |
| air05 | 26374 | 26374 | 26374 | 26439 | 26374 | 26441 | 26374 | 26374 | 26374 | 26374 |
| app1-1 | -3 | -3 | -3 | -3 | -2 | -3 | -3 | -3 | -3 | -3 |
| app1-2 | -23 | -41 | -41 | -24 | -23 | -29 | -41 | -41 | -41 | -41 |
| assign1-5-8 | 212 | 212 | 212 | 214 | 212 | 212 | 212 | 212 | 212 | 212 |
| atlanta-ip | 98.01 | 93.01 | 93.01 | 93.01 | 95.01 | 93.01 | 90.01 | 90.01 | 90.01 | 90.01 |
| b1c1s1 | 27031.16 | 27004.55 | 25571.02 | 27540.75 | 27031.16 | 26379.4 | 24544.25 | 24544.25 | 24544.25 | 24544.25 |
| bab2 | -354064.7 | -354064.7 | -354064.7 | -354064.7 | -354064.7 | -354091 | -354092.9 | -354092.9 | -357525.96 | -357544.312 |
| bab6 | -279121.2 | -279121.2 | -279121.2 | -280546.4 | -280546.4 | -280546.4 | -280546.4 | -280546.4 | -284248.23 | -284248.23 |
| beasleyC3 | 754 | 754 | 754 | 759 | 759 | 755 | 754 | 754 | 754 | 754 |
| binkar10_1 | 6742.2 | 6742.2 | 6742.2 | 6746.76 | 6747.78 | 6743.24 | 6742.2 | 6742.2 | 6742.2 | 6742.2 |
| blp-ar98 | 6565.99 | 6303.11 | 6243.77 | 6565.99 | 6584.43 | 6565.99 | 6205.21 | 6205.21 | 6205.21 | 6205.21 |
| blp-ic98 | 4744.08 | 4719.1 | 4641.77 | 4719.1 | 4963.66 | 4963.66 | 4491.45 | 4491.45 | 4491.45 | 4491.45 |
| bnatt400 | 1 | 1 | 1 | 1 | 1 | 1 | 1 | 1 | 1 | 1 |
| bppc4-08 | 53 | 53 | 53 | 56 | 56 | 54 | 53 | 53 | 53 | 53 |
| brazil3 | 102 | 102 | 102 | 102 | 102 | 102 | 41 | 41 | 24 | 24 |
| buildingenergy | 42652.34 | 34250.38 | 34250.38 | 42652.34 | 42652.34 | 42652.34 | 34243.89 | 34250.38 | 33283.85 | 33283.85 |
| cbs-cta | 0 | 0 | 0 | 43.16 | 43.16 | 0 | 0 | 0 | 0 | 0 |
| chromaticindex1024-7 | 4 | 4 | 4 | 4 | 4 | 4 | 4 | 4 | 4 | 4 |
| chromaticindex512-7 | 4 | 4 | 4 | 4 | 4 | 4 | 4 | 4 | 4 | 4 |
| cmflsp50-24-8-8 | 57921400 | 57921400 | 57921400 | 57921400 | 57921400 | 57921400 | 55789390 | 55789390 | 55789390 | 55789390 |
| CMS750_4 | 261 | 254 | 252 | 261 | 269 | 253 | 252 | 252 | 252 | 252 |
| co-100 | 11833720 | 11833720 | 11833720 | 11833720 | 11833720 | 11833720 | 2639942.06 | 2639942.06 | 2639942.06 | 2639942.06 |
| cod105 | -12 | -12 | -12 | -11 | -11 | -8 | -12 | -12 | -12 | -12 |
| comp07-2idx | 823 | 148 | 148 | 148 | 148 | 78 | 6 | 23 | 6 | 6 |
| comp21-2idx | 250 | 179 | 142 | 250 | 250 | 225 | 75 | 88 | 88 | 74 |
| cost266-UUE | 25222800 | 25148941 | 25148941 | 25222800 | 25222800 | 25164070 | 25148941 | 25148941 | 25148941 | 25148941 |
| csched007 | 362 | 351 | 351 | 362 | 356 | 354 | 351 | 351 | 351 | 351 |
| csched008 | 173 | 173 | 173 | 176 | 178 | 174 | 173 | 173 | 173 | 173 |
| cvs16r128-89 | -93 | -95 | -95 | -86 | -80 | -84 | -97 | -97 | -96 | -97 |
| dano3_3 | 576.345 | 576.345 | 576.345 | 577.475 | 576.52 | 576.52 | 576.345 | 576.345 | 576.345 | 576.345 |
| dano3_5 | 576.925 | 576.925 | 576.925 | 581.725 | 581.725 | 577.316 | 576.925 | 576.925 | 576.925 | 576.925 |
| decomp2 | -160 | -160 | -160 | -152 | -152 | -133 | -160 | -160 | -160 | -160 |
| drayage-100-23 | 103334 | 103334 | 103334 | 103334 | 103334 | 103334 | 103334 | 103334 | 103334 | 103333.874 |
| drayage-25-23 | 101283 | 101283 | 101283 | 106897 | 101344 | 101344 | 101283 | 101283 | 101283 | 101282.647 |
| dws008-01 | 56691.23 | 46179.85 | 38873.46 | 56691.23 | 56691.23 | 56691.23 | 37412.6 | 37412.6 | 37412.6 | 37412.6 |

| Instance | SCIP | SCIP(600s) | SCIP(900s) | U-LNS | R-LNS | FT-LNS | BTBS-LNS | BTBS-LNS-F | Gurobi | Best Solution |
|---|---|---|---|---|---|---|---|---|---|---|
| eil33-2 | 934.008 | 934.008 | 934.008 | 987.674 | 987.674 | 934.008 | 934.008 | 934.008 | 934.008 | 934.008 |
| eilA101-2 | 1313.47 | 995.77 | 995.77 | 1443.53 | 1443.53 | 1443.53 | 880.92 | 880.92 | 923.01 | 880.92 |
| enlight_hard | 37 | 37 | 37 | 37 | 37 | 37 | 37 | 37 | 37 | 37 |
| ex10 | 100 | 100 | 100 | 100 | 100 | 100 | 100 | 100 | 100 | 100 |
| ex9 | 81 | 81 | 81 | 81 | 81 | 81 | 81 | 81 | 81 | 81 |
| exp-1-500-5-5 | 65887 | 65887 | 65887 | 65887 | 65887 | 65887 | 65887 | 65887 | 65887 | 65887 |
| fast0507 | 174 | 174 | 174 | 176 | 175 | 174 | 174 | 174 | 174 | 174 |
| fastxgemm-n2r6s0t2 | 230 | 230 | 230 | 230 | 236 | 230 | 230 | 230 | 230 | 230 |
| fhnw-binpack4-48 | 0 | 0 | 0 | 0 | 0 | 0 | 0 | 0 | 0 | 0 |
| fiball | 140 | 138 | 138 | 138 | 140 | 138 | 138 | 138 | 138 | 138 |
| gen-ip002 | -4783.73 | -4783.73 | -4783.73 | -4772.6 | -4772.6 | -4768.25 | -4783.73 | -4772.33 | -4783.73 | -4783.73 |
| gen-ip054 | 6858.88 | 6840.97 | 6840.97 | 6858.88 | 6852.73 | 6858.88 | 6852.73 | 6858.58 | 6840.97 | 6840.97 |
| germanrr | 48440630 | 48096190 | 48096190 | 48440630 | 48440630 | 48440630 | 48440630 | 48440630 | 47135500 | 47095869.6 |
| glass-sc | 23 | 23 | 23 | 25 | 25 | 24 | 23 | 23 | 23 | 23 |
| glass4 | 1200012600 | 1200012600 | 1200012600 | 1200012600 | 1200012600 | 1200012600 | 1200012600 | 1200012600 | 1200012600 | 1200012600 |
| gmu-35-40 | -2406458 | -2406458 | -2406458 | -2406458 | -2406458 | -2406458 | -2406458 | -2406458 | -2406733 | -2406733.37 |
| gmu-35-50 | -2606871 | -2606930 | -2606930 | -2605465 | -2605387 | -2606930 | -2607958.3 | -2607958.3 | -2607922.7 | -2607958.33 |
| graph20-20-1rand | -9 | -9 | -9 | -8 | -8 | -9 | -9 | -9 | -9 | -9 |
| graphdraw-domain | 19686 | 19686 | 19686 | 19848 | 19848 | 19772 | 19686 | 19688 | 19686 | 19686 |
| h80x6320d | 6382.1 | 6382.1 | 6382.1 | 6416.61 | 6382.1 | 6382.1 | 6382.1 | 6382.1 | 6382.1 | 6382.1 |
| hypothyroid-k1 | -2851 | -2851 | -2851 | -2851 | -2851 | -2851 | -2851 | -2851 | -2851 | -2851 |
| ic97_potential | 3945 | 3945 | 3945 | 3952 | 3945 | 3952 | 3942 | 3952 | 3942 | 3942 |
| icir97_tension | 6392 | 6382 | 6375 | 6375 | 6382 | 6376 | 6375 | 6375 | 6375 | 6375 |
| irp | 12159.49 | 12159.49 | 12159.49 | 12160.2 | 12161.5 | 12161.5 | 12159.49 | 12159.49 | 12159.49 | 12159.49 |
| istanbul-no-cutoff | 204.08 | 204.08 | 204.08 | 214.797 | 212.961 | 214.797 | 204.08 | 204.08 | 204.08 | 204.08 |
| k1mushroom | -204 | -293 | -3288 | -204 | -204 | -204 | -3144 | -3144 | -3288 | -3288 |
| lectsched-5-obj | 48 | 41 | 39 | 46 | 48 | 44 | 24 | 27 | 24 | 24 |
| leo1 | 419655200 | 412600400 | 410709200 | 488216000 | 443395100 | 429182100 | 404989400 | 404989400 | 404227536 | 404227536 |
| leo2 | 436700200 | 426090600 | 424958900 | 438115100 | 436700200 | 436444000 | 405531200 | 405531200 | 404077441 | 404077441 |
| lotsize | 1557868 | 1484323 | 1483960 | 1626587 | 1557868 | 1495682 | 1480195 | 1480195 | 1494101 | 1480195 |
| mad | 0.067 | 0.038 | 0.0352 | 0.067 | 0.0772 | 0.0392 | 0.0268 | 0.0268 | 0.028 | 0.0268 |
| map10 | -480 | -495 | -495 | -468 | -410 | -472 | -495 | -495 | -495 | -495 |
| map16715-04 | -78 | -109 | -111 | -82 | -78 | -83 | -111 | -111 | -111 | -111 |
| markshare_4_0 | 1 | 1 | 1 | 3 | 1 | 1 | 1 | 1 | 1 | 1 |
| markshare2 | 31 | 28 | 28 | 31 | 36 | 31 | 11 | 11 | 11 | 1 |
| mas74 | 11801.19 | 11801.19 | 11801.19 | 11801.19 | 11801.19 | 11801.19 | 11801.19 | 11801.19 | 11801.19 | 11801.1857 |
| mas76 | 40005.05 | 40005.05 | 40005.05 | 40005.05 | 40005.05 | 40005.05 | 40005.05 | 40005.05 | 40005.05 | 40005.05 |
| mc11 | 11689 | 11689 | 11689 | 11720 | 11731 | 11896 | 11689 | 11689 | 11689 | 11689 |
| mcsched | 211913 | 211913 | 211913 | 212874 | 212874 | 212911 | 211913 | 211913 | 211913 | 211913 |
| mik-250-20-75-4 | -52301 | -52301 | -52301 | -52301 | -52301 | -52301 | -52301 | -52301 | -52301 | -52301 |
| milo-v12-6-r2-40-1 | 326481.1 | 326481.1 | 326481.1 | 326820.6 | 326820.6 | 326481.1 | 326481.1 | 326481.1 | 326481.1 | 326481.1 |
| momentum1 | 372399.4 | 282447.1 | 134897 | 365944 | 365944 | 372399.4 | 109143.5 | 109143.5 | 109143.5 | 109143.5 |
| mushroom-best | 0.0553 | 0.0553 | 0.0553 | 0.0869 | 0.0869 | 0.0553 | 0.0553 | 0.0553 | 0.0553 | 0.0553 |
| mzzv11 | -21718 | -21718 | -21718 | -21678 | -21668 | -21678 | -21718 | -21718 | -21718 | -21718 |
| mzzv42z | -20540 | -20540 | -20540 | -20540 | -20540 | -20400 | -20540 | -20540 | -20540 | -20540 |

| Instance | SCIP | SCIP(600s) | SCIP(900s) | U-LNS | R-LNS | FT-LNS | BTBS-LNS | BTBS-LNS-F | Gurobi | Best Solution |
|---|---|---|---|---|---|---|---|---|---|---|
| n2seq36q | 52600 | 52200 | 52200 | 52800 | 52600 | 52400 | 52200 | 52200 | 52200 | 52200 |
| n3div36 | 130800 | 130800 | 130800 | 130800 | 131400 | 130800 | 130800 | 130800 | 130800 | 130800 |
| n5-3 | 8105 | 8105 | 8105 | 8405 | 8105 | 8105 | 8105 | 8105 | 8105 | 8105 |
| neos-1122047 | 161 | 161 | 161 | 161 | 161 | 161 | 161 | 161 | 161 | 161 |
| neos-1171448 | -309 | -309 | -309 | -307 | -305 | -309 | -309 | -309 | -309 | -309 |
| neos-1171737 | -190 | -192 | -192 | -173 | -190 | -190 | -195 | -195 | -195 | -195 |
| neos-1445765 | -17783 | -17783 | -17783 | -17783 | -17783 | -17783 | -17783 | -17783 | -17783 | -17783 |
| neos-1456979 | 186 | 184 | 184 | 207 | 184 | 186 | 176 | 178 | 176 | 176 |
| neos-1582420 | 91 | 91 | 91 | 91 | 91 | 91 | 91 | 91 | 91 | 91 |
| neos-2657525-crna | 7.23 | 7.23 | 7.23 | 7.23 | 8.06 | 7.23 | 1.81075 | 7.23 | 1.81075 | 1.81075 |
| neos-2746589-doon | 2099.6 | 2099.6 | 2099.6 | 2099.6 | 2099.6 | 2099.6 | 2099.6 | 2099.6 | 2008.2 | 2008.2 |
| neos-2978193-inde | -2.388 | -2.388 | -2.388 | -2.197 | -2.388 | -2.388 | -2.388 | -2.388 | -2.388 | -2.3880616 |
| neos-2987310-joes | -607702988 | -607702988 | -607702988 | -607702988 | -607702988 | -607702988 | -607702988 | -607702988 | -607702988 | -607702988 |
| neos-3004026-krka | 0 | 0 | 0 | 0 | 0 | 0 | 0 | 0 | 0 | 0 |
| neos-3024952-loue | 126520 | 97446 | 71336 | 81469 | 97446 | 126520 | 26756 | 27349 | 26756 | 26756 |
| neos-3046615-murg | 1610 | 1610 | 1607 | 1670 | 1651 | 1651 | 1610 | 1611 | 1600 | 1600 |
| neos-3083819-nubu | 6307996 | 6307996 | 6307996 | 6307996 | 6307996 | 6307996 | 6307996 | 6307996 | 6307996 | 6307996 |
| neos-3216931-puriri | 151160 | 151160 | 151160 | 141275 | 151160 | 141275 | 141275 | 141275 | 71320 | 71320 |
| neos-3381206-awhea | 453 | 453 | 453 | 453 | 454 | 453 | 453 | 454 | 453 | 453 |
| neos-3402294-bobin | 0.06725 | 0.06725 | 0.06725 | 0.08775 | 0.06725 | 0.08175 | 0.06725 | 0.06725 | 0.06725 | 0.06725 |
| neos-3555904-turama | -34.7 | -34.7 | -34.7 | -34.7 | -34.7 | -34.7 | -34.7 | -34.7 | -34.7 | -34.7 |
| neos-3627168-kasai | 989301.6 | 989301.6 | 989301.6 | 990006.8 | 989301.6 | 989301.6 | 988585.62 | 988585.62 | 988585.62 | 988585.62 |
| neos-3656078-kumeu | -11067.1 | -11067.1 | -11067.1 | -11067.1 | -11067.1 | -11067.1 | -13127 | -13120 | -13171 | -13172.2 |
| neos-3754480-nidda | 13832.17 | 13639.97 | 13639.97 | 13832.17 | 13832.17 | 13832.17 | 12940.5 | 12940.5 | 12941.69 | 12940.5 |
| neos-4300652-rahue | 7.4454 | 2.8193 | 2.7595 | 6.1813 | 7.4454 | 6.1813 | 2.1416 | 2.1416 | 2.1416 | 2.1416 |
| neos-4338804-snowy | 1477 | 1474 | 1473 | 1482 | 1479 | 1479 | 1471 | 1473 | 1471 | 1471 |
| neos-4387871-tavua | 35.14 | 35.14 | 35.14 | 35.14 | 35.14 | 35.14 | 33.38 | 33.38 | 33.38 | 33.38 |
| neos-4413714-turia | 45.37 | 45.37 | 45.37 | 51.94 | 51.94 | 45.37 | 45.37 | 45.37 | 45.37 | 45.37 |
| neos-4647030-tutaki | 27268.48 | 27268.48 | 27268.48 | 27268.48 | 27268.48 | 27268.48 | 27265.71 | 27265.71 | 27265.71 | 27265.71 |
| neos-4722843-widden | 25438.44 | 25210.88 | 25210.88 | 27707.88 | 26277.44 | 26277.44 | 25009.7 | 25309.66 | 25009.7 | 25009.7 |
| neos-4738912-atrato | 285010500 | 283680800 | 283680100 | 285662900 | 285662900 | 285010500 | 283676100 | 283627957 | 283627957 | 283627957 |
| neos-4763324-toguru | 6760.735 | 6760.735 | 6760.735 | 6760.735 | 6760.735 | 6760.735 | 1613.039 | 1613.039 | 1613.039 | 1613.039 |
| neos-4954672-berkel | 2678506 | 2627560 | 2624735 | 2678506 | 2678506 | 2678506 | 2612710 | 2612710 | 2614881 | 2612710 |
| neos-5049753-cuanza | 636 | 636 | 636 | 636 | 636 | 600 | 600 | 600 | 562 | 562 |
| neos-5052403-cygnet | 293 | 293 | 184 | 293 | 293 | 293 | 182 | 182 | 182 | 182 |
| neos-5093327-huahum | 6686 | 6686 | 6686 | 6686 | 6960 | 6686 | 6260 | 6260 | 6270 | 6260 |
| neos-5107597-kakapo | 4248 | 3744 | 3654 | 4194 | 4293 | 4158 | 3645 | 3645 | 3645 | 3645 |
| neos-5188808-nattai | 0.11257 | 0.11257 | 0.11207 | 0.11257 | 0.11257 | 0.11257 | 0.11029 | 0.11029 | 0.11029 | 0.11029 |
| neos-5195221-niemur | 0.00406 | 0.00384 | 0.00384 | 0.00418 | 0.00418 | 0.00406 | 0.00384 | 0.00384 | 0.00384 | 0.00384 |
| neos-631710 | 214 | 214 | 214 | 214 | 214 | 214 | 203 | 203 | 203 | 203 |
| neos-662469 | 245034.5 | 184745.5 | 184679.5 | 224993.5 | 225044 | 245034.5 | 184380 | 184390 | 184380 | 184380 |
| neos-787933 | 30 | 30 | 30 | 30 | 30 | 30 | 30 | 30 | 30 | 30 |
| neos-827175 | 112.002 | 112.002 | 112.002 | 112.002 | 112.002 | 112.002 | 112.002 | 112.002 | 112.002 | 112.002 |
| neos-848589 | 12359660 | 2359.54 | 2359.54 | 12359660 | 12359660 | 12359660 | 2358.843 | 2358.43 | 3206.12 | 2351.4031 |
| neos-860300 | 3201 | 3201 | 3201 | 3201 | 3267 | 3201 | 3201 | 3201 | 3201 | 3201 |

| Instance | SCIP | SCIP(600s) | SCIP(900s) | U-LNS | R-LNS | FT-LNS | BTBS-LNS | BTBS-LNS-F | Gurobi | Best Solution |
|---|---|---|---|---|---|---|---|---|---|---|
| neos-873061 | 122.92 | 122.72 | 122.72 | 125.93 | 122.92 | 123.66 | 113.656 | 113.656 | 113.656 | 113.656 |
| neos-911970 | 54.76 | 54.76 | 54.76 | 54.83 | 54.83 | 54.76 | 54.76 | 54.76 | 54.76 | 54.76 |
| neos-933966 | 2388 | 320 | 320 | 2389 | 2388 | 2388 | 318 | 318 | 318 | 318 |
| neos-950242 | 4 | 4 | 4 | 4 | 5 | 4 | 4 | 4 | 4 | 4 |
| neos-957323 | -237.76 | -237.76 | -237.76 | -234.76 | -235.76 | -235.76 | -237.76 | -237.76 | -237.76 | -237.76 |
| neos-960392 | 0 | -238 | -238 | 0 | 0 | -234 | -238 | -238 | -238 | -238 |
| neos17 | 0.15 | 0.15 | 0.15 | 0.171 | 0.167 | 0.151 | 0.15 | 0.15 | 0.15 | 0.15 |
| neos5 | 15 | 15 | 15 | 15 | 15 | 15 | 15 | 15 | 15 | 15 |
| neos8 | -3719 | -3719 | -3719 | -3719 | -3719 | -3719 | -3719 | -3719 | -3719 | -3719 |
| net12 | 214 | 214 | 214 | 214 | 255 | 214 | 214 | 214 | 214 | 214 |
| netdiversion | 4900438 | 4900438 | 263 | 4900438 | 4900438 | 263 | 244 | 244 | 242 | 242 |
| nexp-150-20-8-5 | 300 | 234 | 231 | 771 | 771 | 237 | 231 | 231 | 239 | 231 |
| ns1208400 | 2 | 2 | 2 | 2 | 2 | 2 | 2 | 2 | 2 | 2 |
| ns1644855 | -1419.67 | -1524.33 | -1524.33 | -1486.67 | -1486.67 | -1419.67 | -1524.33 | -1524.33 | -1524.33 | -1524.33 |
| ns1760995 | -429.36 | -429.36 | -429.36 | -429.36 | -429.36 | -429.36 | -548.02 | -548.02 | -516.07 | -549.214385 |
| ns1830653 | 20622 | 20622 | 20622 | 23622 | 23622 | 21622 | 20622 | 20622 | 20622 | 20622 |
| nu25-pr12 | 53905 | 53905 | 53905 | 53905 | 53905 | 53905 | 53905 | 53905 | 53905 | 53905 |
| nursesched-medium-hint03 | 8080 | 8080 | 7906 | 8080 | 8080 | 8080 | 117 | 997 | 152 | 115 |
| nursesched-sprint02 | 58 | 58 | 58 | 58 | 67 | 58 | 58 | 58 | 58 | 58 |
| nw04 | 16862 | 16862 | 16862 | 16876 | 16876 | 16876 | 16862 | 16862 | 16862 | 16862 |
| opm2-z10-s4 | -29112 | -33062 | -33062 | -26538 | -26538 | -26538 | -33269 | -33269 | -33139 | -33269 |
| p200x1188c | 15078 | 15078 | 15078 | 15078 | 15078 | 15078 | 15078 | 15078 | 15078 | 15078 |
| pg | -8674.34 | -8674.34 | -8674.34 | -8662.84 | -8662.84 | -8674.34 | -8674.34 | -8674.34 | -8674.34 | -8674.34 |
| pg5_34 | -14324.46 | -14324.81 | -14325.83 | -14310.96 | -14324.46 | -14324.81 | -14339.4 | -14339.4 | -14339.4 | -14339.4 |
| physiciansched6-2 | 49324 | 49324 | 49324 | 49324 | 49324 | 49324 | 49324 | 49324 | 49324 | 49324 |
| piperout-08 | 125055 | 125055 | 125055 | 133707 | 125055 | 125055 | 125055 | 125055 | 125055 | 125055 |
| piperout-27 | 8124 | 8124 | 8124 | 8124 | 8124 | 8124 | 8124 | 8124 | 8124 | 8124 |
| pk1 | 11 | 11 | 11 | 12 | 12 | 11 | 11 | 11 | 11 | 11 |
| proteindesign121hz512p9 | 2609 | 2609 | 2609 | 2609 | 2609 | 2609 | 2609 | 2609 | 1477 | 1473 |
| proteindesign122trx11p8 | 2916 | 2916 | 2916 | 2916 | 2916 | 2916 | 1767 | 1762 | 1748 | 1747 |
| qap10 | 340 | 340 | 340 | 340 | 340 | 340 | 340 | 340 | 340 | 340 |
| radiationm18-12-05 | 19527 | 18874 | 18874 | 19853 | 19527 | 19202 | 17566 | 17569 | 17567 | 17566 |
| radiationm40-10-02 | 235396 | 155354 | 155354 | 235396 | 235396 | 209796 | 155330 | 156939 | 155331 | 155328 |
| rail01 | -69.09 | -69.09 | -69.09 | -69.89 | -69.89 | -69.09 | -69.89 | -69.89 | -70.57 | -70.57 |
| rail507 | 174 | 174 | 174 | 178 | 180 | 175 | 174 | 174 | 174 | 174 |
| ran14x18-disj-8 | 3715 | 3714 | 3712 | 3798 | 3798 | 3715 | 3712 | 3712 | 3736 | 3712 |
| rd-rplusc-21 | 179751.8 | 179751.8 | 179751.8 | 179836.5 | 179751.8 | 179751.8 | 165395.3 | 165395.3 | 165395.3 | 165395.3 |
| reblock115 | -36721080 | -36799530 | -36800600 | -36777270 | -36799530 | -36799530 | -36800603 | -36800603 | -36800603 | -36800603 |
| rmatr100-p10 | 423 | 423 | 423 | 442 | 424 | 457 | 423 | 423 | 423 | 423 |
| rmatr200-p5 | 5489 | 5489 | 4521 | 5489 | 5489 | 5489 | 4521 | 4521 | 4521 | 4521 |
| rocI-4-11 | -6020203 | -6020203 | -6020203 | -5040303 | -5040303 | -6020203 | -6020203 | -6020203 | -6020203 | -6020203 |
| rocII-5-11 | -4.65 | -5.66 | -5.67 | -4.65 | -5.66 | -4.65 | -6.68 | -6.68 | -5.68 | -6.68 |
| rococoB10-011000 | 19988 | 19879 | 19534 | 19701 | 19879 | 19988 | 19449 | 19449 | 19497 | 19449 |
| rococoC10-001000 | 11530 | 11460 | 11460 | 11576 | 11472 | 11460 | 11460 | 11460 | 11460 | 11460 |
| roi2alpha3n4 | -61.37 | -63.17 | -63.17 | -62.41 | -62.41 | -63.17 | -63.21 | -63.21 | -63.21 | -63.21 |

| Instance | SCIP | SCIP(600s) | SCIP(900s) | U-LNS | R-LNS | FT-LNS | BTBS-LNS | BTBS-LNS-F | Gurobi | Best Solution |
|---|---|---|---|---|---|---|---|---|---|---|
| roi5alpha10n8 | -44.89 | -45.15 | -45.15 | -44.36 | -44.89 | -44.89 | -52.28 | -52.28 | -50.59 | -52.3222744 |
| roll3000 | 12890 | 12890 | 12890 | 12902 | 12890 | 12890 | 12890 | 12890 | 12890 | 12890 |
| s100 | 0 | 0 | 0 | 0 | 0 | 0 | -0.16966 | -0.16966 | -0.03945 | -0.16972527 |
| s250r10 | -0.1437 | -0.1698 | -0.1708 | -0.1437 | -0.1437 | -0.1437 | -0.17178 | -0.17178 | -0.17178 | -0.17178 |
| satellites2-40 | 49 | 49 | 49 | 49 | 49 | 49 | -19 | -19 | -19 | -19 |
| satellites2-60-fs | 28 | 28 | 27 | 27 | 27 | 27 | -19 | -19 | -19 | -19 |
| savsched1 | 31846.3 | 31846.3 | 31846.3 | 45875.9 | 45875.9 | 31846.3 | 3265 | 3265 | 3218 | 3218 |
| sct2 | -230.91 | -230.99 | -230.99 | -230.78 | -230.85 | -230.91 | -230.99 | -230.99 | -230.99 | -230.99 |
| seymour | 427 | 425 | 423 | 427 | 428 | 427 | 423 | 423 | 423 | 423 |
| seymour1 | 410.76 | 410.76 | 410.76 | 410.76 | 410.76 | 410.76 | 410.76 | 410.76 | 410.76 | 410.76 |
| sing326 | 7833336 | 7765711 | 7765711 | 7833336 | 7833336 | 7833336 | 7753675 | 7753675 | 7753676 | 7753675 |
| sing44 | 8175655 | 8174767 | 8174767 | 8177833 | 8163698 | 8175655 | 8128831 | 8128831 | 8130643 | 8128831 |
| snp-02-004-104 | 586912700 | 586816300 | 586804500 | 586829700 | 587089300 | 586821500 | 586804500 | 586803239 | 586803239 | 586803239 |
| sorrell3 | -11 | -15 | -15 | -11 | -15 | -15 | -16 | -16 | -16 | -16 |
| sp150x300d | 69 | 69 | 69 | 69 | 69 | 70 | 69 | 69 | 69 | 69 |
| sp97ar | 688832800 | 682989900 | 681332100 | 681332100 | 673491900 | 679524100 | 660834000 | 660834000 | 660705646 | 660705646 |
| sp98ar | 537245600 | 533010800 | 532905600 | 533010800 | 533455300 | 532891300 | 529905800 | 529905800 | 529740623 | 529740623 |
| splice1k1 | -73 | -121 | -394 | -121 | -121 | -121 | -394 | -394 | -338 | -394 |
| square41 | 26 | 26 | 26 | 26 | 21 | 21 | 15 | 17 | 16 | 15 |
| square47 | 29 | 29 | 29 | 21 | 21 | 21 | 18 | 20 | 20 | 16 |
| supportcase10 | 19 | 19 | 19 | 9 | 19 | 19 | 8 | 8 | 8 | 7 |
| supportcase12 | -7430.15 | -7437.1 | -7475.67 | -7351.97 | -7436.17 | -7449.13 | -7543.26 | -7543.26 | -7559.2419 | -7559.2419 |
| supportcase18 | 49 | 49 | 49 | 50 | 51 | 49 | 48 | 48 | 49 | 48 |
| supportcase26 | 1781.003 | 1747.033 | 1747.033 | 1755.845 | 1768.264 | 1768.264 | 1755.525 | 1755.525 | 1745.124 | 1745.124 |
| supportcase33 | -345 | -345 | -345 | -340 | -345 | -345 | -345 | -345 | -345 | -345 |
| supportcase40 | 24478.86 | 24465.78 | 24465.78 | 24465.78 | 24478.86 | 24294.09 | 24256.31 | 24256.31 | 24256.31 | 24256.31 |
| supportcase42 | 8.0904 | 8.0019 | 7.7683 | 7.7678 | 7.7685 | 7.7811 | 7.7678 | 7.7713 | 7.7586 | 7.7586 |
| supportcase6 | 51921.76 | 51921.76 | 51921.76 | 51921.76 | 51921.76 | 51906.48 | 51906.48 | 51906.48 | 51906.48 | 51906.48 |
| supportcase7 | -1132.223 | -1132.223 | -1132.223 | -1129.28 | -1132.223 | -1132.223 | -1132.223 | -1132.223 | -1132.223 | -1132.223 |
| swath1 | 379.07 | 379.07 | 379.07 | 379.07 | 381.51 | 379.07 | 379.07 | 379.07 | 379.07 | 379.07 |
| swath3 | 397.76 | 397.76 | 397.76 | 399.33 | 397.76 | 397.76 | 397.76 | 397.76 | 397.76 | 397.76 |
| tbfp-network | 131.88 | 24.16 | 24.16 | 25.12 | 24.91 | 24.16 | 24.16 | 24.16 | 24.16 | 24.16 |
| thor50dday | 59310 | 59310 | 40432 | 40432 | 40432 | 40432 | 40417 | 40417 | 40417 | 40417 |
| timtab1 | 764772 | 764772 | 764772 | 766166 | 766345 | 764772 | 764772 | 764772 | 764772 | 764772 |
| tr12-30 | 130596 | 130596 | 130596 | 130608 | 130596 | 130608 | 130596 | 130596 | 130596 | 130596 |
| traininstance2 | 79180 | 77420 | 77420 | 79180 | 84090 | 79180 | 72030 | 72950 | 71820 | 71820 |
| traininstance6 | 29420 | 28460 | 28460 | 28290 | 28460 | 29250 | 28290 | 29250 | 28290 | 28290 |
| trento1 | 25255630 | 18223810 | 18223810 | 18223810 | 7282245 | 15981790 | 5191562 | 5191562 | 5189487 | 5189487 |
| triptim1 | 25.5 | 25.5 | 25.5 | 22.87 | 25.5 | 22.87 | 22.87 | 22.87 | 22.87 | 22.87 |
| uccase12 | 11507.41 | 11507.41 | 11507.41 | 11507.42 | 11507.48 | 11507.41 | 11507.41 | 11507.41 | 11507.41 | 11507.41 |
| uccase9 | 463233.3 | 48328.09 | 15347.75 | 48328.09 | 48328.09 | 20176.81 | 11052.31 | 11052.31 | 10994.13 | 10993.1314 |
| uct-subprob | 315 | 314 | 314 | 317 | 315 | 314 | 314 | 314 | 314 | 314 |
| unical_7 | 19635620 | 19635558 | 19635558 | 19635558 | 19635558 | 19635558 | 19635558 | 19635558 | 19635558 | 19635558 |
| var-smallemery-m6j6 | -149.375 | -149.375 | -149.375 | -147.031 | -146.312 | -149.375 | -149.375 | -149.375 | -149.375 | -149.375 |
| wachplan | -8 | -8 | -8 | -8 | -8 | -8 | -8 | -8 | -8 | -8 |

