# OpenReview forum: "BTBS-LNS: A Binarized-Tightening, Branch and Search Approach of Learning Large Neighborhood Search Policies for MIP"
_ICLR.cc/2024/Conference — ICLR 2024 Conference Withdrawn Submission_

### Official Review · Reviewer_uy5T · 2023-10-27

**Soundness:** 4 excellent
**Presentation:** 1 poor
**Contribution:** 4 excellent
**Rating:** 6
**Confidence:** 3

**Summary:**

In this paper, the authors develop a machine-learning-guided large neighborhood search (LNS) procedure that provides a heuristic approach to general MIPs. This includes several improvements over existing (LNS) procedures including:
- A new approach to handling general binary variables, wherein a learning model is used to predict the tightness of bounds placed on these variables
- A new way of encoding MIP problems as graphs, wherein the graph includes nodes for the objective function
- A method wherein a learning model is used to generate local branching constraints
The authors provide extensive computational results on a range of problems.

**Strengths:**

The paper features some novel improvements to machine-learning-guided large neighborhood search approaches to MIP. This is a timely topic with high potential significance. The computational results are extensive and impressive.

**Weaknesses:**

Overall, I found this paper to be extremely unclear and poorly written. For example:
- The authors state that "we represent each general integer variable with $d=\lceil \log_2(ub - lb) \rceil$ variables, where ub and lb are original variable upper and lower bounds, respectively. The subsequent optimization is applied to the substitute variables". The authors should show this substitution more explicitly. The authors seem to indicate that the bound-tightening scheme is then fixing some of these substitute variables. I am aware of some common binarization schemes (e.g. Owen & Mehrota 2002), but the bound-tightening scheme given in algorithm 1 does not seem to correspond with a variable-fixing approach to any binarization scheme that I can think of.
-  I was completely unable to understand even at a high-level what their "branching" approach entailed until I read Algorithm 3 in the appendix. Readers should not have to read the appendix to make sense of the main text. In fact, their algorithm does not really "branch" as far as I can tell. "Branching" typically means building a search tree that divides the space of solutions. Instead, the algorithm adds local branching constraints similar to Fischetti and Lodi (2003) or Liu, Fischetti and Lodi (2022), but doesn't actually carry out the branching part of this procedure. Note: the authors might disagree with the statement "the algorithm adds local branching constraints", as the authors differentiate between "global branching" and "local branching", but in either case the constraint that is added takes the form $\sum\_{i \in D} |x^{t+1}\_i - x_i^t| \leq rn$, which is a local branching constraint as defined in Fischetti and Lodi 2003.
- In order to understand how the authors defined a neighborhood for their large neighborhood search procedure, I again had to consult the appendix. This is a critical detail of a large neighborhood search algorithm, and should be made much more clear.
- I cannot tell what data was used to train these models in any of the experiments.

The three improvements (bound tightening scheme, graph encoding & learning approach, "branching" learning approach) in the paper each seem somewhat marginal. However, I don't view this weakness as something that would necessarily prevent publication of this work, as the combination of these improvements does seem to be a significant advancement in the design of heuristic procedures for MIPs.

**Questions:**

What binarization scheme did you use?
What training data did you use in your computational experiments?

---

> ### Author Response · Authors · 2023-11-15
> **Response to Reviewer uy5T**
>
> We sincerely thank Reviewer uy5T for the positive feedback and precious suggestions! Below we address each comment in detail:
>
> > **Q1**: Detail explanation on the "Binarize and Tighten" approach.
>
> Thanks for your comments. For general integer variables, we propose a "Binarize and Tighten" scheme. At first, in the binarization phase, general integer variables were encoded to $d = \lceil{\log_{2}{(ub-lb)}}\rceil$ substitution variables. For example, a general integer variable $x$ with range $[0, 7]$ can be encoded to 3 variables ($x_1, x_2, x_3$) to represent the original variable with decreasing significance. In this way, we transform the LNS decision for the original variable to 3 different decisions on substitution variables. Then, when either variable is fixed by LNS, it indicates current solution reliable at corresponding significance. The decisions are utilized to tighten the bounds of original variables sequentially, rather than directly fixed, at current step illustrated in Alg.1.
>
>
> > **Q2**: Explanation on the branching components, as it doesn't actually carry out the branching part of this procedure.
>
> Thanks for your insightful comments. The meaning of branching in this paper mainly refers to the one used in the previous work [1]. In other words, we identify the potentially wrongly decided variables by LNS and optimize them by adding the local branching constraints. It takes effect on top of LNS, making them closer to global optimum and serves as a further optimization scheme to the learned LNS policy.
>
>
> > **Q3**: In order to understand how the authors defined a neighborhood for their large neighborhood search procedure, I again had to consult the appendix. This is a critical detail of a large neighborhood search algorithm and should be made much more clear.
>
> Thanks for your suggestions. We will update the paper and place some critical descriptions back to the main text if page permits.
>
>
>
> > **Q4**: I cannot tell what data was used to train these models in any of the experiments.
>
> As illustrated in Sec 4.1, for SC, MIS, CA and MC, we generate several training instances following the same distribution with the testing set, except for different problem sizes. Then the LNS and branching policy will be sequentially trained on these instances. The features and states for the instances are given in Table 7 in the appendix.
>
> As for heterogeneous instances like MIPLIB, we performed cross-validation. Specifically, at each round, we split them into training, validation, and testing sets by 70\%, 15\%, and 15\%. In this way, the whole dataset can be tested sequentially to make fair comparison.
>
>
> **References:**
>
> [1] Liu D, Fischetti M, Lodi A. Learning to search in local branching[C]//Proceedings of the AAAI Conference on Artificial Intelligence. 2022, 36(4): 3796-3803.

---

> > ### Comment · Reviewer_uy5T · 2023-11-20
> > **Response to author response**
> >
> > Thank you for the clarifications. I think that the changes have made many things about the paper more clear, and I would revise my overall rating to an 8 (accept).

---

> > > ### Author Response · Authors · 2023-11-22
> > > **Response to reviewer uy5T**
> > >
> > > Thanks for the recognition to our work, and look forward your raising the rating score as you mentioned. We will continue to revise and improve the paper to meet the highest standard.

---

### Official Review · Reviewer_zCZB · 2023-10-29

**Soundness:** 2 fair
**Presentation:** 3 good
**Contribution:** 2 fair
**Rating:** 5
**Confidence:** 4

**Summary:**

In this paper, the authors propose the BTBS-LNS technique to solve the MIP problem in order to cope with the problem that the LNS technique often falls into local optimality. The authors claim that their technique effectively escapes local optimality, and extensive experiments on a large number of instances show that it leads the SCIP and LNS baseline, is comparable to Gurobi, and even performs better early in the run.

**Strengths:**

1. The article is praiseworthy for its extensive experimental data and significant findings. The authors have selected a large number of baselines for comparative experiments on different benchmarks, which are demonstrated by a large number of figures and tables. These experimental results strongly prove the effectiveness of the new technology.

2. The paper is laudable for its well-structured and logical presentation, providing a comprehensive understanding of the research topic.

**Weaknesses:**

1. On page 4, in ‘THE BINARIZED TIGHTENING SCHEME’ chapter, the authors mention that the analysis for MIPLIB shows that all unbounded variables have either upper or lower bounds, and the authors have therefore designed virtual upper/lower bounds. Could there be meaningful MIP instances other than MIPLIB where variables exist simultaneously with no upper or lower bounds? If it exists, can BTBS-LNS handle it? Are there serious robustness problems with ignoring such instances?

2. On page 8, in 'Hyperparameters' under section 4.1, the authors mention the use of SCIP as the default solver, noting that this is the blue box part of Fig. 1. In my understanding the SCIP solver should correspond to the 'MIP solver' on the right hand side of Fig.1, and the boxes for Neighborhood Search and Branching Policy on the left hand side are similarly blue, is this a minor graphing and writing error, or am I misunderstanding that SCIP is equally involved in these stages?

3. As described in the experiment part, the authors conduct experiments on those MIP instances that are encoded from other combinatorial problems, including set covering, maximal independent set, combinatorial auction, and maximum cut. Actually, there are specific optimization solvers for each of those combinatorial optimization problems, and MIP solvers do not represent the state of the art in solving those problems. As a submission to a top-tier conference, it is required to compare their proposed method against the real state of the art.

4. In fact, there exist standard benchmarks for evaluating MIP solvers, i.e., Hans Mittelmann's benchmarks (https://plato.asu.edu/bench.html). Why don’t the authors test their proposed method and baseline solvers on those standard benchmarks?

**Questions:**

Please see my comments in "Weaknesses".

**Details Of Ethics Concerns:**

I do not have ethic concern about this paper.

---

> ### Author Response · Authors · 2023-11-15
> **Response to reviewer zCZB**
>
> We sincerely thank Reviewer zCZB for their valuable suggestions. Summary of responses and revisions are as follows:
>
> > **Q1**: Could there be meaningful MIP instances other than MIPLIB where variables exist simultaneously with no upper or lower bounds? If it exists, can BTBS-LNS handle it? Are there serious robustness problems with ignoring such instances?
>
> We really appreciate your insightful comments. Although rare, there may be instances that contain variables simultaneously with no upper or lower bounds (called "free" variables). In our current implementation, we will leave them unfixed (untightened). We argue that it was not serious robustness problems, as the presolve technique in MIP solvers can handle most of the cases into single unbounded or totally bounded. The few remaining variables being unfixed will not contribute to heavy computational costs. To be more convincing, we will test our approach on those instances to further evaluate its generalization ability, in our next round of response with updated experiments.
>
>
> >**Q2**: On page 8, in 'Hyperparameters' under section 4.1, the authors mention the use of SCIP as the default solver, noting that this is the blue box part of Fig.1. In my understanding the SCIP solver should correspond to the 'MIP solver' on the right hand side of Fig.1, and the boxes for Neighborhood Search and Branching Policy on the left hand side are similarly blue, is this a minor graphing and writing error, or am I misunderstanding that SCIP is equally involved in these stages?
>
> Sorry for some misunderstanding. SCIP only corresponds to the 'MIP solver' on the right hand side of Fig.1. We will make it clear in the updated paper version.
>
>
> > **Q3**: As described in the experiment part, the authors conduct experiments on those MIP instances that are encoded from other combinatorial problems, including set covering, maximal independent set, combinatorial auction, and maximum cut. Actually, there are specific optimization solvers for each of those combinatorial optimization problems, and MIP solvers do not represent the state of the art in solving those problems. As a submission to a top-tier conference, it is required to compare their proposed method against the real state of the art.
>
>
> Thanks for your insightful suggestions. Similar to the MIP solvers, our proposed approach was designed for general instances, rather than specific problems. We don't finetune our structures or frameworks specifically for each problem. In this respect, except for the specific problems, we also compared our BTBS-LNS on more heterogeneous instances (MIPLIB) from diverse problems, and it also delivers superior performance compared with the MIP solvers, further demonstrating the generalization ability. However, we really appreciate your valuable suggestions. Comparison with specific optimization solvers may be an interesting topic to further improve and demonstrate the performance of our proposed BTBS-LNS, and we will try to supplement some experiments, in our next round of response with updated experiments.
>
>
> > **Q4**: In fact, there exist standard benchmarks for evaluating MIP solvers, i.e., Hans Mittelmann's benchmarks (https://plato.asu.edu/bench.html). Why don’t the authors test their proposed method and baseline solvers on those standard benchmarks?
>
> Thanks for your comments. We have evaluated our proposed approaches on these standard benchmarks. The anonymous MIPLIB (AMIPLIB) tested in Table 3 contains of a curated set of instances from MIPLIB, which are benchmark instances to test MIP solvers from Hans Mittelmann (https://plato.asu.edu/ftp/milp.html). And to make full comparison, we have also tested the whole 240 benchmark instances from MIPLIB in Appendix A.5, and detailed performance analysis are reported in Appendix B. In general, our proposed BTBS-LNS can achieve competitive performance even on these standard cross-distribution instances.

---

### Official Review · Reviewer_38AJ · 2023-10-30

**Soundness:** 2 fair
**Presentation:** 1 poor
**Contribution:** 3 good
**Rating:** 5
**Confidence:** 4

**Summary:**

A deep learning-based approach for providing primal solutions to mixed-integer programming (MIP) problems is described. The primary contributions are given as follows. First, the paper proposes "binarized tightening" for variable encoding / adjusting bounds during search. Second, a tripartite graph model is proposed in which the "standard" bipartite graph model is extended with nodes representing the objective function. Third, the paper uses a large neighborhood search fix-and-resolve strategy and claims this is a variable branching strategy. Fourth, they compare the approach to SCIP and Gurobi on the several MIP datasets and claim good performance.

Overall, this is a paper that unfortunately is not good enough for acceptance. This is especially disappointing because it has some fascinating ideas; I think both the tripartite graph and the bound tightening, if properly explained, could be important contributions.

=====
I have raised my score from a 3 to a 5. See my comment for more information.

**Strengths:**

1. The tripartite graph structure is an interesting idea. One might argue that the objective information was already present in the graph, so this isn't necessary to add in this way, however I think the objective nodes could provide an interesting flow of information between parts of the problem that are connected through objective components.

2. The binarized tightening is indeed new for the learning to optimize literature. It reminds me of tightening schemes in constraint programming, so it would be nice if the authors could make that connection in their work. There is an imitation learning aspect to this that I am not sure is really so new, but in general the encoding of the variables is interesting.

3. The performance of the approach is quite promising.

**Weaknesses:**

1. This paper is not well written and unfortunately this results in a variety of issues. Please have a professional proofreader check for typos, there are several including in the abstract (e.g., there must be a "the" before "Mixed").

a. Key contributions of the paper are barely even discussed in the main text. Information about the branch and search method, tripartite graph, etc., is all in the back. And these are not just details, in my opinion. This paper basically abuses the page limit with lots of redundant text (BTBS is defined several times...) and then expects the reviewer to read the appendix to actually understand what is going on. Sorry, I am not willing to do that. Furthermore, the bound tightening scheme seems somewhat arbitrary -- there are many tightening/filtering strategies, see the CP literature.

b. There are experiments at the start of the paper before the reader can even really understand what is being tested, on what instances, etc. This is essentially a visualization for marketing purposes rather than science.

c. The conclusion is essentially non-existent, nor do the authors discuss weaknesses of their work.


2. I take great issue with the big claim in the abstract that the approach performs competitively with Gurobi and SCIP. As best I can tell, this approach is a heuristic, meaning this is an apples and oranges comparison. If this approach actually finds optimal solutions and provides a proof, then I truly have no idea how, which only furthers my first argument.

3. The paper makes several claims/arguments that I just do not follow/support.

a. Backdoor variables: I see no theoretical or empirical proof that this paper finds them. It is just a conjecture that it is finding backdoors; we do not really know. The seminal paper on backdoors from Williams, Gomes and Selman ("Backdoors to typical case complexity") is not even cited.

b. The paper argues that current approaches "becom[e] ensnared in local optima". The implication is that this method does not, but of course, it does. This is a sentence that can be written about essentially any heuristic method and is thus not a valid argument for this paper about why it is novel or better than anything else. Table 1 continues this misconception and basically gets everything wrong about metaheuristics and "addressing local optima". Succinctly put, LNS itself is a high level strategy for avoiding local optima, the adaptive neighborhood size of other approaches is also a mechanism for doing so. The header just makes no sense for categorizing the techniques.

In the example with (3), I do not view this issue as a fundamental flaw of LNS. LNS is just a framework that can be implemented however the user likes. They are correct that fixing integer variables to specific values in their domains is ineffective (again, see the past 20 years of the constraint programming literature), but this has nothing to do with LNS as a framework.

4. The experiments are all given 200 seconds for solving. What a surprise that SCIP and Gurobi do not solve many instances that well in that time frame. It is totally unreasonable and completely disconnected with the time limits that one would have when solving a general MIP problem. Generally ten minutes is considered a reasonable amount of time for a heuristic on a large-scale real-world problem. For solving to optimality with SCIP or Gurobi, one would usually allow for 24 (or more) hours.

Furthermore, the experiments do not provide sensitivity analysis or ablation in the main text beyond comparing different branching schemes. Maybe there is more in the appendix, but those are critical details for the main text.

**Questions:**

1. Can this method prove optimality?
2. Is there an inherent reason why the bound tightening works in powers of two?
3. Why do you claim that MIPs are more difficult than IPs? Please cite the CS theory that says this is the case.

---

> ### Author Response · Authors · 2023-11-15
> **Response to the reviewer 38AJ (Part 1)**
>
> We thank reviewer 38AJ so much for the careful review and insightful comments! Below we would like to illustrate some details.
>
> > **Q1**: This paper is not well written and results in a variety of issues and typos.
>
> Thanks for your suggestions. We will check the overall expressions and writing in detail and correct some typos in the updated paper version.
>
> > **Q2**: Key contributions of the paper are barely even discussed in the main text. Furthermore, the bound tightening scheme seems somewhat arbitrary -- there are many tightening/filtering strategies, see the CP literature.
>
> Thanks for your valuable suggestions. We will place some important components, e.g., branch and search method, tripartite graph, back to the main text if page permits.
>
> As for the bound tightening scheme, it was designed for general integer variables. The intuition is to balance between complexity and exploration scope. In this respect, we designed a binarize-then-tighten scheme. The binarization was utilized to encode them with several substitute decision variables, utilized to evaluate the reliability of current solution at different significance. And afterwards, the bitwise LNS decisions were applied to the bound tightening sequentially. When either bit indicates the current solution reliable, we tighten the bounds that maintains a pivotal position close to (or precisely at) the midpoint of the recalibrated bounds. We argue that when the current solution sits precisely at the midpoint of variable bounds, no further bound tightening should be performed as it shows no preference for either direction, which drives us to design the current bound tightening scheme, tightening the bounds on the far side iteratively.
>
> > **Q3**: There are experiments at the start of the paper before the reader can even really understand what is being tested, on what instances, etc. This is essentially a visualization for marketing purposes rather than science.
>
> We really appreciate your insightful comments. The existence of Fig.2 was utilized to intuitively illustrate the potential limitations and drawbacks of current learning based LNS approaches on instances from Maxcut and MIPLIB. However, it may be confusing for the readers to some extent in the methodology section. It may be better to illustrate the limitations in a clear way.
>
> > **Q4**: The conclusion is essentially non-existent, nor do the authors discuss weaknesses of their work.
>
> Thanks for your suggestions, and currently the conclusion and weaknesses given in Sec 5 were not sufficient. In general, we have proposed a binarized tightening branch and search approach to learn LNS policies in this paper. It was designed to efficiently deal with general MIP problems, and delivers superior performance over competing baselines on ILP, MIP datasets and even heterogeneous instances from MIPLIB. Sufficient ablation studies demonstrate the effectiveness of each component, including the tripartite graph, binarize and tighten scheme, and the extra branching at each step.
>
> However, the proposed BTBS-LNS is only a primal heuristic to search for better feasible solutions, while cannot prove optimality, which are also common limitations of LNS-based approaches. Implementing them into MIP solvers as primal heuristics may be a possible solution. However, interaction with current existed primal heuristics, and the rule to take effect, are key challenges in practical implementation. In general, applications of the learning-based approach in real-world scenarios will be our future directions.
>
> > **Q5**: I take great issue with the big claim in the abstract that the approach performs competitively with Gurobi and SCIP. As best I can tell, this approach is a heuristic, meaning this is an apples and oranges comparison.
>
> Thanks for your valuable comments. Our proposed approach is an improved primal heuristic to search for better feasible solutions. Similar to all the heuristic based LNS methods, it cannot prove optimality. As you mentioned, MIP solvers, like Gurobi and SCIP, will simultaneously find feasible solutions and prove the optimality. To make fair comparison, both Gurobi and SCIP were fine-tuned with the aggressive mode to focus on improving the objective value, rather than proving optimality. It follows the same protocol as previous publications [1, 2, 3]. Compared with the hybrid multiple primal heuristics implemented in MIP solvers, our proposed BTBS-LNS can deliver better solutions within the same timelimit, demonstrating its effectiveness.
>
> **Note that owing to the limited characters of only one response and we want to answer your questions in detail, we have to add another reply in the following!**

---

> ### Author Response · Authors · 2023-11-15
> **Response to the reviewer 38AJ (Part 2)**
>
> **Owing to the limited characters of only one response and we want to answer your questions in detail, we have to add another reply in the following!**
>
> > **Q6**: The paper makes several claims/arguments that I just do not follow/support, e.g., backdoor variables.
>
> Thanks for your insightful comments. Backdoor variables refer to a set of variables, such that when set correctly, the sub-solver can solve the remaining problem [4]. LNS may benefit from the backdoor variables due to its partial fixing mechanism. If we can find the backdoors and fix them correctly as soon as possible, the LNS may deliver significantly better performance. In this paper, we extend the meaning of backdoor variables to those with different solutions compared with global optimum (**BTBS-LNS-G**), which may be confusing to some extent. We have clarified its meaning in the abstract. To identify and optimize those variables efficiently, we propose an extra branching policy on top of LNS at each step.
>
> > **Q7**: The paper argues that current approaches "become ensnared in local optima", which has to be explained
>
> We really appreciate your comments, and we will update some expressions to make it clear. As you mentioned, the descriptions for LNS and local optimum may be inaccurate. In fact, the drawbacks and limitations of current learning based LNS approaches may derive from limited learning accuracy or performance. Driven by the learning complexity of MIP problems, current approaches may be easy to be ensnared in local optima. It has nothing to do with LNS as a framework. In other words, **what we claimed "ensnared in local optima" totally refers to the non-optimal learning based LNS policies, rather than the LNS itself**. Hybrid policies or adaptive neighborhood size were both strategies to deal with the limitations of learning-based approaches. Specifically for our BTBS-LNS, it can be regarded as an improved LNS approach by identifying and optimizing the potentially wrongly decisions by the learned LNS policy. And therefore, it can help the LNS decisions getting closer to global optimum, jumping out of local optimum in some cases.
>
> > **Q8**: The experiments are all given 200 seconds for solving, which may be unreasonable.
>
> Thanks for your comments. As illustrated above, our BTBS-LNS was an improved primal heuristic to search for better feasible solutions, while not prove optimality. In MIP solvers, multiple primal heuristics are called iteratively at each branch and bound node, with strong time/iteration/node constraints to ensure efficiency. They try to find feasible solutions of good quality in a reasonably short period of time [5]. And to make fair comparison, both Gurobi and SCIP was fine-tuned with the aggressive mode to focus on improving the objective value, rather than proving optimality. The time limit 200s was set following similar protocols with the previous publication [1], and within 200s, some approaches can already deliver quite close gaps compared with the global optimum on most of the instances. To make comparisons in detail, anytime performance within 200s timelimit was reported in Fig.4 and Fig.6, 7, 8, 9 in the appendix, revealing the superior performance over competing baselines almost anytime from 0s to 200s.
>
> > **Q9**: The experiments do not provide sensitivity analysis or ablation in the main text beyond comparing different branching schemes. Maybe there is more in the appendix, but those are critical details for the main text.
>
> Thanks for your valuable suggestions. In Sec 4.4, we make sensitivity analysis on the branching variable ratios. And detailed sensitivity analysis on other important hyperparameters will be supplemented in our next round of response.
>
> As for the ablation study, we tested the degraded versions of our BTBS-LNS by removing each component separately, including
> - **LNS-TG**: where we replace the tripartite graph with the widely used bipartite graph.
> - **LNS-Branch**: where we remove the branching policy.
> - **LNS-IBT**, **LNS-IT**: where we remove the binarize & tighten mechanism separately.
> - **LNS-ATT**: where we replace our attention-based graph attention network with the widely used GNN.
>
> As can be seen from Table 1, 2, 3, they all performed slightly worse, revealing the effectiveness and necessity of each component.
>
> **Note that owing to the limited characters of only one response and we want to answer your questions in detail, we have to add another reply in the following!**

---

> ### Author Response · Authors · 2023-11-15
> **Response to the reviewer 38AJ (Part 3)**
>
> **Owing to the limited characters of only one response and we want to answer your questions in detail, we have to add another reply in the following!**
>
> > **Q10**: Is there an inherent reason why the bound tightening works in powers of two?
>
> We really appreciate your valuable comments. Intuitively, we utilized a similar binarization encoding scheme to efficiently control the encoded variable counts, avoiding heavy computational costs. And on the other hand, considering that the general integer variables may deliver different ranges, an adaptive bound tightening scheme may be more suitable to deal with the dynamic ranges across variables. With the current scheme, each encoded variable can represent the original integer variable at different significance, utilized to sequentially tighten the bounds.
>
> > **Q11**: Why do you claim that MIPs are more difficult than IPs? Please cite the CS theory that says this is the case.
>
> Sorry for some confusing expressions, and we will update them to make it clear. In this paper, the difficulty of MIP compared with IPs mainly lie in that the continuous variables may require different treatment in learning-based approaches, as they can rarely be fixed. While in general, the difficulty of both MIP and IPs can vary depending on the characteristics of the specific problems being considered. Different problem instances may present unique challenges that impact the complexity and feasibility of finding optimal solutions.
>
> **References**:
>
> [1] Wu, Yaoxin, et al. "Learning large neighborhood search policy for integer programming." Advances in Neural Information Processing Systems 34 (2021): 30075-30087.
>
> [2] Song J, Yue Y, Dilkina B. A general large neighborhood search framework for solving integer linear programs[J]. Advances in Neural Information Processing Systems, 2020, 33: 20012-20023.
>
> [3] Huang T, Ferber A M, Tian Y, et al. Searching large neighborhoods for integer linear programs with contrastive learning[C]//International Conference on Machine Learning. PMLR, 2023: 13869-13890.
>
> [4] Williams R, Gomes C P, Selman B. Backdoors to typical case complexity[C]//IJCAI. 2003, 3: 1173-1178.
>
> [5] Achterberg T. Constraint integer programming[J]. 2007.

---

> > ### Comment · Reviewer_38AJ · 2023-11-21
> >
> > I appreciate the effort the authors have taken to try to respond to the paper's weaknesses, but let me first emphasize that I believe the comment field has a character limit for a good reason.
> >
> > There is no revision process here, and I cannot possibly re-read the paper to see if it is actually well-written now. The best I can offer is that I have looked through it, and it honestly does not look that changed. The contributions at the beginning are a great change, but you still mix experiments into the beginning of the paper.
> >
> > I am not satisfied with the connection to constraint programming here; bounds tightening was not invented in your paper, and I see no connection to the existing literature.
> >
> > I am willing to increase my score to a 5, acknowledging some improvements. But I am not an advocate for this paper at this time, sorry. I believe you have some nice ideas in here, and I am confident that when you have the time to really re-work the paper you will have a very successful submission.

---

> > > ### Author Response · Authors · 2023-11-22
> > > **Further response to Reviewer 38AJ**
> > >
> > > Thank you for your continued feedback which is really inspirational. We have carefully considered your comments and made additional revisions to the paper, including the removal of Figure 2 and the addition of a discussion on various bound tightening techniques in the field of constraint programming. These changes are marked in blue in the latest updated version.
> > >
> > > > Q1: The contributions at the beginning are a great change, but you still mix experiments into the beginning of the paper.
> > >
> > > Thank you for your comments. We understand the importance of clarity and readability in the paper. Therefore, we have decided to remove Figure 2 from the updated version to ensure a clearer separation between the introduction of our contributions and the presentation of experimental details.
> > >
> > >
> > > > Q2: I am not satisfied with the connection to constraint programming here.
> > >
> > > We genuinely appreciate your valuable suggestions. As you mentioned, bound tightening has been applied in various constraint integer programming and mixed integer programming problems. In response, we have elaborated on several common bound tightening techniques and clarified their connections to our proposed scheme.
> > >
> > > Our revised version now includes a discussion on the following bound tightening techniques:
> > >
> > > + **Domain (Constraint) Propagation**: This technique involves tightening the domains of variables by inspecting constraints and the current domains of other variables [1]. It encompasses methods such as Linear Constraint Propagation [2] and set covering constraint propagation (equivalent to SAT clauses) [3]. Domain propagation aims to propagate variable domains within individual constraints, resulting in tightened bounds.
> > >
> > >
> > > + **Optimization-Based Bound Tightening (OBBT) [4]**. OBBT operates at the problem level and considers all constraints collectively. It formulates maximum/minimum optimization problems for each variable, incorporating all constraints to derive their tightest upper/lower bounds.
> > >
> > > + **Feasibility-based bound tightening (FBBT)** This technique leverages primal feasibility arguments to eliminate parts of the variable domains that do not contain feasible solutions [5].
> > >
> > > We acknowledge that these techniques have been implemented in solvers like Gurobi and SCIP, and they share similar insights with our approach in terms of reducing the solving complexity of the re-defined problem. These techniques aim to maintain optimality, yet making them sometimes computationally expensive. In contrast, our iterative refinement procedure for bound tightening differs from them. Our decisions on bound tightening depend on the current solution and the learned LNS policy, as outlined in Alg. 1. The iterative optimization scheme focuses on searching for better feasible solutions within the neighborhood of the current solution, guided by the learned policy. Consequently, our approach allows for a significant reduction of the complexity of the re-defined problem, leading to improved solutions efficiently.
> > >
> > > We sincerely appreciate your thorough evaluation and constructive feedback. Your insights have been invaluable in guiding our revisions. We are committed to addressing your concerns and further improving the paper to meet the highest standards.
> > >
> > > Finally we are open for your further comments and questions. Thanks!
> > >
> > >
> > > **References**:
> > >
> > > [1] Achterberg T. Constraint integer programming[J]. 2007.
> > >
> > > [2] Savelsbergh M W P. Preprocessing and probing techniques for mixed integer programming problems[J]. ORSA Journal on Computing, 1994, 6(4): 445-454.
> > >
> > > [3] Moskewicz M W, Madigan C F, Zhao Y, et al. Chaff: Engineering an efficient SAT solver[C]//Proceedings of the 38th annual Design Automation Conference. 2001: 530-535.
> > >
> > > [4] Gleixner A M, Berthold T, Müller B, et al. Three enhancements for optimization-based bound tightening[J]. Journal of Global Optimization, 2017, 67: 731-757.
> > >
> > > [5] Belotti P, Cafieri S, Lee J, et al. On feasibility based bounds tightening[J]. 2012.

---

### Official Review · Reviewer_5yM1 · 2023-11-01

**Soundness:** 2 fair
**Presentation:** 1 poor
**Contribution:** 3 good
**Rating:** 5
**Confidence:** 5

**Summary:**

The paper presents an ML-guided LNS framework for MIPs. It combines a binarize-and-tighten scheme to address general integer variables and a branching policy to help escape local optima. It also has a search policy to guide destroy variables. In the experiment, the new method is compared against a variety of ML-guided approaches and heuristic approaches. The presented results show that the proposed method finds better solutions at a faster speed.

**Strengths:**

1. The paper proposes a combination of ML-guided search policy and branching policy for LNS in MIP solving. Various techniques and engineering designs are proposed. The novelty of the paper comes from those details.

2. The effectiveness is demonstrated in experiments with different settings and ablation studies.

**Weaknesses:**

1. The writing for some important parts is difficult to understand. I don’t follow the argument for the BTBS scheme. Though intuitively the method itself makes sense to me. In figure 1, are the neighborhood search and branching policy two separate local searches in one iteration? From the pseudocode in Appendix, it looks like you destroy and repair the variables you got by taking the intersection of the branching and LNS policies.  But in other places, you said the branching policy is on top of the LNS.

2. Details on how you implement the branching policy are totally missing, e.g., how you collect data and train. The variant BTBS-LNS-L looks very similar to Sonnerat et al. The design for the branching policy is not well-justified conceptually.

3. Minor weakness: The experiment is comprehensive with lots of information, but without a summary it makes readers get lost. It would be nice to summarize the takeaways, especially for those ablation studies.

**Questions:**

See the weaknesses.

---

> ### Author Response · Authors · 2023-11-15
> **Response to the reviewer 5yM1 (Part 1)**
>
> We sincerely thank Reviewer 5yM1 for the detailed feedback and precious suggestions! Below we address every comment in detail.
>
> > **Q1**: The writing for some important parts is difficult to understand. In Fig.1, are the neighborhood search and branching policy two separate local searches in one iteration?
>
> Thanks for your comments. The large neighborhood search (LNS) and branching are two separate learned policies. At each iteration, current problem states will be firstly fed into the learned LNS policy network, and obtain the fix/unfix decisions, denoted as $n_i^t$. Afterwards, the neighborhood search decisions, along with the problem states, will serve as inputs for the branching networks (see Sec 3.4 for detail), and the branching decisions for the potentially wrongly fixed variables by LNS can be obtained, denoted as $b_i^t$. In this respect, branching policy are indeed on top of LNS to optimize some potentially wrong LNS decisions. Then as illustrated in Alg.2, $n_i^t$ and $b_i^t$ are utilized to jointly optimize the current solution.
>
> > **Q2**: Details on how you implement the branching policy are totally missing, e.g., how you collect data and train. The variant BTBS-LNS-L looks very similar to Sonnerat et al. The design for the branching policy is not well-justified conceptually.
>
> Thanks for your valuable suggestions. Training pipeline for the branching policy was briefly introduced in Alg.3 in the appendix. Specifically, we propose to learn the branching policy with an offline manner. The inputs are tripartite graph-based features (same as LNS policy learning), where we additionally append the LNS decisions at each step as variable features, as we only focused on the fixed variables for extra branching. In this respect, branching can be seen on top of LNS, which was significantly different from Sonnerat et al [1]. Then the graph-based features are fed into a similar graph attention network as described in Sec 3.3 to update the node/edge representations. We finally process the variable nodes by a multi-layer perceptron to obtain the branching probability for each variable at this step. Cross-entropy loss was utilized to train the branching network to bring the outputs closer to the collected labels
>
> Specifically for the data collection, with the learned LNS policy, we solve the training instances for each problem again. The tripartite graph-based features were collected at each step, along with the LNS decisions. Label collection were different between the two branching variants. In the local branching, labels are collected by incorporating the following constraint at each step:
>
> $$\sum\limits_{i \in {\mathcal B} \cap {\mathcal F}} {|x_i^{t + 1} - x_i^t| \le k}$$
>
> where ${\mathcal F}$ denotes the fixed variables selected by LNS. With this extra constraint, the re-defined sub-MIP can be solved, and up to $k$ branching variables are selected from ${\mathcal F}$, serving as branching labels. In the global branching variant, it gathers labels also from the fixed variables by LNS at each step and contrast them with the global optimal solution. Variables that exhibit differing values are indicative of potentially misclassified variables within the current LNS decisions.
>
> In general, the intuition of the branching component is to select and optimize those wrongly fixed variables by the learned LNS policy at each step. The global and local branching variants are designed at a global (local) view, to identify the variables that sit far from the global optimum and optimize them with an extra branching process. To further evaluate its effectiveness, we compared our BTBS-LNS with LNS-Branch, where we remove the extra branching policy at each step. Below are the results on complex MIP problems (item placement and AMIPLIB), and the full comparison results can refer to Table 1, 2, and 3.
>
> + **Balanced item placement**
>
> |Methods|Obj|Gap%|Primal Integral|
> |:--:|:--:|:--:|:--:|
> |LNS-Branch|20.12|43.90|3537.0|
> |BTBS-LNS-L|13.82|16.82|2030.3|
> |BTBS-LNS-G|**13.45**|**15.78**|**1912.5**|
>
> + **Anonymous MIPLIB (AMIPLIB)**
>
> |Methods|Gap%|
> |:--:|:--:|
> |LNS-Branch|9.32|
> |BTBS-LNS-L|**4.19**|
> |BTBS-LNS-G|4.35|
>
> As can be seen, our BTBS-LNS achieves consistently and significantly superior performance over LNS-Branch, revealing the necessity of extra branching on top of LNS.
>
> **Note that owing to the limited characters of only one response and we want to answer your questions in detail, we have to add another reply in the following!**

---

> > ### Author Response · Authors · 2023-11-15
> > **Response to the reviewer 5yM1 (Part 2)**
> >
> > **Owing to the limited characters of only one response and we want to answer your questions in detail, we have to add another reply in the following!**
> >
> > > **Q3**: The experiment is comprehensive with lots of information, but without a summary it makes readers get lost. It would be nice to summarize the takeaways, especially for those ablation studies.
> >
> > Thanks for your suggestions. In this paper, we compared our BTBS-LNS on seven different datasets, including ILP, MIP and standard MIP benchmarks (MIPLIB), which contained instances from diverse problem distributions. The competing baselines contained SOTA MIP solvers (SCIP and Gurobi), common heuristic-based LNS implemented in MIP solvers (RENS, RINS, ...), and learning-based LNS policies (RL and Imitation learning). In addition, some degraded versions of BTBS-LNS were also considered as competing baselines to illustrate the effectiveness of each component, including
> > - **LNS-TG**: where we replace the tripartite graph with the widely used bipartite graph.
> > - **LNS-Branch**: where we remove the branching policy.
> > - **LNS-IBT**, **LNS-IT**: where we remove the binarize & tighten mechanism separately.
> > - **LNS-ATT**: where we replace our attention-based graph attention network with the widely used GNN.
> >
> > The overall results indicate the necessity of each component, demonstrating the effectiveness of our hybrid branch and search framework, even on extremely difficult instances. Our approach can perform competitively with, sometimes even better, than Gurobi (fine-tuned to aggressive mode that focused on improving the objective value) in finding better feasible solutions.
> >
> > In addition, we also evaluated our approach on its anytime performance (not a single 200s timelimit, see Appendix A.4), stability (see Appendix A.8), and GPU **vs** CPU analysis, further illustrating its effectiveness and availability.
> >
> >
> > **References**:
> >
> > [1] Sonnerat et al. “Learning a large neighborhood search algorithm for mixed integer programs.” arXiv preprint arXiv:2107.10201 (2021).

---

### Author Response · Authors · 2023-11-15
**Global response on the revisions of the paper**

We sincerely thank all the reviewers for their feedback and constructive comments, which help us significantly improve the paper quality. Below we would like to illustrate some improvements of the updated paper version (marked in blue in the paper):

- Rearrangement of the paper
    - Place Table 1 on the comparison with related works to Appendix A.1.
    - Place some less important experiments (Section 4.5 with Gurobi as baseline solver) to Appendix A.6.
    - Place the algorithm on the hybrid branch and search pipeline back to the main text, to make it easy to understand.
- Update some expressions in the abstract and clarify the concept for backdoor variables in the footnotes of the abstract.
- Update Fig.1 to make it clear (color for some boxes).
- Illustrate some details for "Binarize and Tighten scheme";
- Make Alg.2 clear that the branching and LNS are two separate learned policies.
- Illustrate the design intuition, training and data collection of the branching network in Sec 3.4;
- Correct some issues and typos.
- Summarize the paper and illustrate the weaknesses in Sec 5.

---

### Author Response · Authors · 2023-11-21
**Generel response on further discussion**

Dear AC and reviewers,

We would like to express our sincere gratitude again for your valuable comments and thoughtful suggestions. Throughout the rebuttal phase, we tried our best to address the concerns and refine details in alignment with your constructive feedback. All the changes are listed in the former global response, and also marked in blue in the updated paper version. In addition, we have carefully addressed each specific concerns and questions in our individual response to each reviewer.

We really thank the reviewer uy5T for the recognition to our work, and look forward your raising the rating score as you mentioned.

**To the reviewer 5yM1, 38AJ and zCZB**

Since the discussion time window is very tight and is approaching its end, we truly hope that our responses have met your expectations and assuaged any concerns. We genuinely do not want to miss the opportunity to engage in further discussions with you, which we hope could contribute to a more comprehensive evaluation of our work. Should any lingering questions persist, we are more than willing to offer any necessary clarifications.

**To AC**

Thanks for your time and efforts hosting the review of our paper. Look forward your helping organizing the reviewer discussion.


With heartfelt gratitude and warmest regards,

The Authors